# Parallel and convergent genomic changes underlie independent subterranean colonization across beetles

Pau Balart-García[1] ✉, Leandro Aristide[1], Tessa M. Bradford[2,3], Perry G. Beasley-Hall[2,3], Slavko Polak[4], Steven J. B. Cooper [2,3] & Rosa Fernández [1] ✉

Adaptation to life in caves is often accompanied by dramatically convergent changes across distantly related taxa, epitomized by the loss or reduction of eyes and pigmentation. Nevertheless, the genomic underpinnings underlying cave-related phenotypes are largely unexplored from a macroevolutionary perspective. Here we investigate genome-wide gene evolutionary dynamics in three distantly related beetle tribes with at least six instances of independent colonization of subterranean habitats, inhabiting both aquatic and terrestrial underground systems. Our results indicate that remarkable gene repertoire changes mainly driven by gene family expansions occurred prior to underground colonization in the three tribes, suggesting that genomic exaptation may have facilitated a strict subterranean lifestyle parallelly across beetle lineages. The three tribes experienced both parallel and convergent changes in the evolutionary dynamics of their gene repertoires. These findings pave the way towards a deeper understanding of the evolution of the genomic toolkit in hypogean fauna.

Extreme habitat transitions such as life in high mountains, deep oceans and caves provide striking scenarios of convergent evolution at the phenotypic level across distantly related taxa, suggesting that under similar environmental conditions there is a limited number of functionally viable solutions provided by evolution[1,2]. The underground realm is one system in which remarkable phenotypic convergence has occurred[3,4]. Many strictly subterranean species share similar phenotypic changes, mostly associated with 'regressive' characters such as the lack of pigmentation, eye reduction or loss, or modification of the circadian rhythm. These traits have evolved multiple times independently as a result of key selective environmental constraints inherent to the subterranean lifestyle such as the absence of light, climatic stability (i.e., constant temperature and high environmental humidity), and the heterogeneity of nutrients both in space and time[5].

Despite historically attracting the attention of evolutionary biologists due to this striking phenotypic convergence, little is known about the genomic underpinnings of these convergent traits. This is partly due to the difficulty in collecting cave organisms and characterizing the environmental conditions of their habitats[6–9]. Recent studies have provided valuable insights into the molecular processes related to loss of eyes and pigmentation, among other regressive phenotypes, particularly in fish[10–13] and crustaceans[14–16]. The evolution of specific gene families involved in circadian rhythm[17], photoreception[18,19], chemosensation[20] and heat stress tolerance[21] have been recently investigated in cave beetles. Nevertheless, convergent evolution in subterranean animals remains poorly understood at the genomic level, especially from a genome-wide macroevolutionary perspective.

[1]Metazoa Phylogenomics Lab, Biodiversity Program, Institute of Evolutionary Biology (CSIC - Universitat Pompeu Fabra), Passeig Marítim de la Barceloneta 37-49, 08003 Barcelona, Spain. [2]Department of Ecology and Evolutionary Biology, School of Biological Sciences, and Environment Institute, University of Adelaide, Adelaide, SA 5005, Australia. [3]South Australian Museum, Adelaide, SA 5000, Australia. [4]Notranjska Museum Postojna, Kolodvorska c. 3, 6230 Postojna, Slovenia. ✉e-mail: pau.balart@ibe.upf-csic.es; rosa.fernandez@ibe.upf-csic.es

Cave-dwelling beetles represent ideal systems to explore the genomic basis of convergent evolution associated with a strictly subterranean lifestyle. These insects comprise numerous instances of phylogenetically-distinct subterranean lineages with varying degrees of morphological change and ecological specialization related to their adaptation to underground habitats. The tribe Leptodirini (Leiodidae, Cholevinae) is the most extensive animal radiation in subterranean habitats, with about 950 known species which are mostly subterranean[22]. Numerous epigean and subterranean lineages are found in the North of the Mediterranean basin and represent parallel 'ancient' colonizations (*ca.* 33 Mya) and subsequent radiations in terrestrial underground habitats[23]. Diving beetles of the tribes Bidessini and Hydroporini (Dytiscidae, Hydroporinae) also represent large radiations in subterranean aquatic ecosystems; they colonized independently multiple underground desert aquifers in Western Australia *ca.* 7–3 Mya[24,25]. Previous studies revealed that both of these distantly related beetle lineages come from surface-dwelling ancestors likely with cavernicolous or straminicolous habitat preferences[26–29].

Here, we explore the genomic underpinnings of adaptations to life in caves in Coleoptera using a genome-wide phylogenomic approach to investigate and characterize gene repertoire evolutionary dynamics across surface-dwelling and subterranean species that represent multiple independent underground colonizations both in terrestrial (Leptodirini) and aquatic beetle lineages (Hydroporini and Bidessini). These lineages belong to the Polyphaga and Adephaga suborders respectively, diverging more than 300 Mya. Our results indicate that genomic exaptation prior to the colonization of subterranean habitats was key to facilitating these transitions in terrestrial and aquatic lineages, and that both parallel and convergent evolution paved the way towards adapting to life in caves across distantly related Coleoptera.

## Results and discussion

### Genomic exaptation facilitated subterranean colonization across beetle lineages

We inferred orthology relationships among 41 proteomes from high-quality genomic and transcriptomic datasets spanning the main beetle lineages (Myxophaga, Archostemata, Adephaga and Polyphaga) plus two outgroups (Neuroptera and Strepsiptera), in order to explore gene repertoire evolution in a broad phylogenetic framework that encompass the lineages of interest. These datasets contain 22 species from the tribes Leptodirini, Hydroporini, and Bidessini, including 21 newly sequenced transcriptomes for this study that represented 6 independent instances of subterranean colonization, with transitions being older in Leptodirini than in Hydroporini and Bidessini (Fig. 1a). Orthology inference resulted in 14,988 orthogroups (OGs hereafter, referred to as gene families throughout the text) comprising more than 3 sequences. We first explored significant shifts in the net gene family expansion rate (i.e., the difference between gene gain and loss rates, net rate hereafter) in branches where lineages were inferred to have transitioned to underground environments. We compared a global rates (GR) estimation model (i.e., assuming the same net rate for all the branches) to a subterranean rates (SR) model (i.e., assuming that subterranean branches have a different net rate compared to the others) through maximum-likelihood-based phylogenetic comparative methods. We explored these shifts in two independent analyses, either including all taxa or including only the aquatic and the terrestrial clade separately. 9.3% of all OGs (1394 OGs) showed significantly different rates in the SR model, thus indicating that relevant changes occurred therein in the subterranean lineages. The analysis including all taxa revealed significant changes in the evolutionary rate related to gene family contraction and expansion (Fig. 1b), indicating that not only gene loss, but also gene gain and duplication, may have facilitated

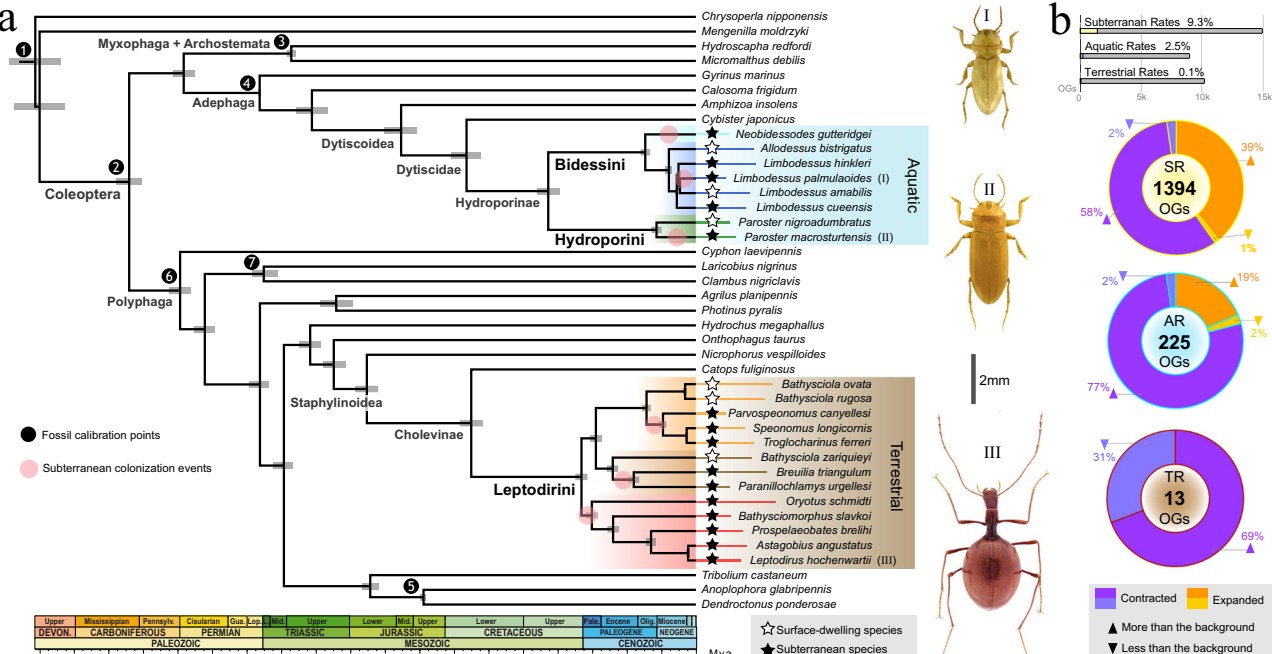

**Fig. 1 | Phylogenetic context and global estimation of gene gain and loss in the transition to subterranean habitats. a** Time-calibrated phylogeny of the studied Coleoptera and outgroups. Black dots indicate the fossil calibration points used to estimate the divergence times (see Supplementary Table 1) and bars on nodes represent the 95% confidence intervals of the estimated split dates. Beetle habitus at the right side of the phylogeny correspond to examples of subterranean species of each tribe included in the study. Photos: © Chris Watts and Howard Hamon (I and II) and Hlaváč et al.[30] (III). **b** Estimation of significant net rate changes in terms of gene gain and loss across all subterranean lineages (subterranean rates model, SR) and at a lineage-specific level (aquatic rates, AR; terrestrial rates, TR). Bar plot indicates the total OGs of each set and the percentage of significant OGs. Pie charts indicate the total number of orthogroups (OGs) with significant changes in the subterranean lineages compared to the background. Source data are provided as a Source Data file and at https://github.com/MetazoaPhylogenomicsLab/Balart_Garcia_et_al_2023_gene_repertoire_evolution_subterranean_coleoptera/blob/main/Source_data_1.zip.

the transition to subterranean environments. Regarding the lineage-specific analyses to explore shifts in gene family net rate in the aquatic or terrestrial lineages, we found that 234 gene families (~2%) showed significant rate changes in the aquatic lineage (with two thirds of the gene families being contracted and one third being expanded), and only 13 in the terrestrial one (all of them contracted) (Fig. 1b). These results have two relevant implications: first, beetles of the aquatic lineage underwent a larger number of changes in their gene repertoire associated with the underground habitat shift compared to the terrestrial ones. Second, the number of gene families with significant changes in net rate is tenfold in the analysis including all species compared to the lineage-specific ones. This indicates that changes in the gene repertoire turnover are only significant if we broaden the phylogenetic context for comparison, and consequently implies that these changes may have occurred prior to the transition to the underground.

To further investigate this we inferred genome-wide gene gain, loss, and duplication in all branches and nodes in the phylogeny based on all OGs. Our results showed that the largest events of gene gain, duplication and loss occurred precisely at the branches leading to the most recent common ancestors (MRCA hereafter) of each tribe (i.e., Bidessini, *ca.* 29.3 Mya; Hydroporini, *ca.* 22.8 Mya; Leptodirini, *ca.* 66.2 Mya), followed by less changes in the subsequent branches (i.e., the ones that lead to the lineages with the independent transitions to life in caves) (Fig. 2a). These findings indicate that notable genomic modifications in the gene repertoire occurred in the ancestors of the lineages that subsequently colonized underground habitats. Furthermore, gene gains were more abundant than losses, suggesting that an increase in the gene repertoire took place before these lineages transitioned to caves (Fig. 2a). The MRCA of Leptodirini showed a larger proportion of contracted OGs compared to the MRCAs of Hydroporini and Bidessini (in any case of smaller magnitude than gene gain and duplication), indicating that gene loss also contributed potentially to pave the way to the underground colonization in the terrestrial tribe but not in the aquatic tribes (Fig. 2a). Remarkably, the consecutive branches representing the three independent transitions to the underground in Leptodirini showed virtually no changes in their gene repertoire dynamics, reinforcing the results described above for the maximum-likelihood phylogenetic comparative methods (i.e., not many significant changes in gene family net rate occurred along the branches associated with transitions to subterranean environments).

Our results thus indicate that the MRCAs of the three tribes experienced a significant level of gene gain and duplication, and suggest that genomic exaptation fueled by an expansion of the gene repertoire may have paved the way for a later colonization of the underground in these three tribes.

## Aquatic and terrestrial beetle lineages experienced both parallel and convergent evolution prior to and during the transition to subterranean environments

To characterize the level of parallel and convergent evolution between lineages with independent transitions to life in caves, we next explored the putative function of the gene families that underwent significant changes in their net rates through an orthology-based functional annotation approach. In the context of this study, we refer to parallel evolution as significant shifts in net rates occurring in gene families shared by all or several lineages. In contrast, we define convergent evolution as lineage-specific significant changes in net rates in gene families that have similar putative functions when compared across lineages[32].

Around 1–5% of the gene families that were expanded, contracted, gained or lost in the node or branch leading to the MRCAs of all tribes were shared by the three of them, ~8–16% were shared by two tribes, and the large majority underwent lineage-specific changes in their net rates (82–86%; Fig. 2c). Gene ontology (GO) enrichment highlighted

the functional similarity between the functions enriched across lineages, which were highly similar in the three MRCAs (Fig. 2b). Noticeably, biological processes related to sensory perception, response to stimulus, development, regulation, morphogenesis and metabolic processes are the most enriched functional categories. Some overrepresented functions appear both in the expanded and contracted OGs, suggesting a complex dynamic and reshaping for some gene families, such as those involved in developmental processes which are expanded and contracted in parallel in the Bidessini and Hydroporini ancestors but only contracted in the tribe Leptodirini. Other enriched functions include locomotion, behavioral and reproductive processes, changes in the cuticle, biomineralization and hydrocarbon metabolic processes, pigmentation, circadian rhythm processes and growth, among other examples. Remarkably a lower metabolic rate, a thinning of the cuticle and changes in its composition, depigmentation, loss of the light dependent regulation of the circadian rhythm, enlargement of body and elongation of sensory appendages have been previously observed to be convergent features of underground species[5, 33]. Our results thus indicate convergent functional changes driven mostly by gene gain and duplication in the MRCAs of the three tribes before the transitions to the underground. These 'new genetic pieces' in the gene repertoire may have provided the substrate facilitating *a posteriori* adaptation to life in caves in independent lineages.

We next investigated the OGs that experienced parallel contractions and expansions when the aquatic and terrestrial lineages shifted to underground habitats in the six independent subterranean colonization events (i.e., shared OGs between two or more lineages; Fig. 3a). Our results suggest that the gene repertoires of the tribes Bidessini and Hydroporini experienced more changes in their gene repertoires during their habitat shift, evidenced by a large number of expanded and contracted OGs in the branches where the subterranean transition occurred, in contrast with the Leptodirini lineages that had a very reduced number of changes (Fig. 3b). The same tendency was found in the number of parallelly expanded and contracted OGs: the aquatic lineages presented a higher degree of parallel evolution (e.g., 114 contracted OGs and 545 expanded OGs shared by B1, B2 and H1), in contrast to the terrestrial lineages that just showed a handful of parallelly expanded and contracted OGs between pairs of lineages. However, a small number of OGs showed parallel evolution across aquatic and terrestrial lineages. In addition, results revealed a higher number of parallelly expanded OGs than those parallelly contracted, suggesting that parallel evolution was mainly driven by gene gain. These results suggest that the transition to caves may have been enabled to some extent by parallel evolution in the three aquatic lineages but not in the terrestrial lineages. We detected a few OGs with parallel changes between aquatic and terrestrial lineages, suggesting that adaptation to the environmental constraints of the subterranean habitats may have been facilitated by parallel genomic changes across distantly related taxa, at least in a few OGs. Regarding the putative function of parallelly expanded or contracted OGs in cave-dwelling lineages, our results indicate that a large proportion of OGs share similar annotations (Fig. 3c). Axon guidance and regulation of transcription are the most represented biological processes both in parallel contractions and expansions. These results show that gene families related to chemotaxis and regulation of gene expression seem to have been shaped both by gene family expansion and contraction in the transition to caves. Furthermore, functions related to neuronal processes also seem to have been deeply modified (i.e., dendrite morphogenesis, axon midline choice point recognition, motor neuron axon guidance, nervous system development, regulation of axonogenesis, axon extension, among other examples), suggesting that expansions in gene families related to the nervous system development and functioning may have been co-opted to functions important to adapt to

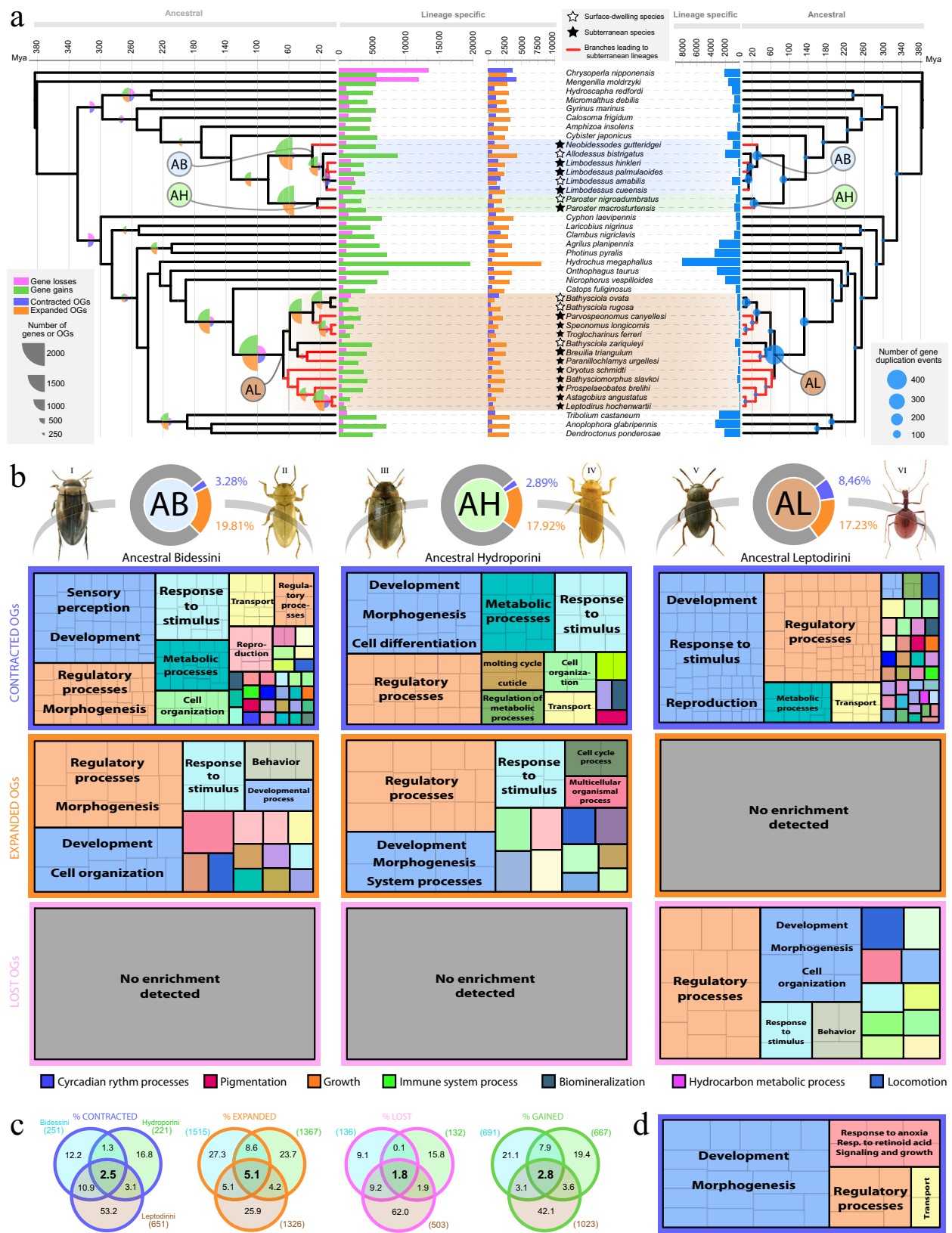

the underground. Remarkably, we found both parallel contraction and expansion of gene families related to compound eye development mainly between aquatic lineages (Fig. 3c). Functional annotation of these contracted gene families indicate that they correspond to basic elements of the eye development in insects (e.g., *MYOSA, RBBP6, OPA1, FZR1*), whereas the expanded gene families are putatively involved in negative regulation of eye development[34] (e.g., *ATXN2*) among other alternative roles in the nervous system development (e.g., *SDK2, boss, EYS*) (Supplementary Data 1). For instance, *EYS* (eyes shut / spam) plays a key role in mechanosensory/chemosensory neurons by preserving cell shape under environmental stress[35] and could be related to extra-optic modifications in

**Fig. 2 | Global reconstruction of the gene repertoire evolution in Coleoptera and enriched functions in the MRCAs of each beetle tribe. a** Symmetric ultrametric trees indicating the number of gene losses and gains, orthogroup contractions and expansions (scaled pie charts in the branches, left) and total gene duplication events (scaled dots in the nodes, right). Colored circles indicate the most recent common ancestor (MRCA) of the tribes Bidessini (AB, Ancestral Bidessini), Hydroporini (AH, Ancestral Hydroporini) and Leptodirini (AL, Ancestral Leptodirini). **b** Pie charts indicate the percentage of contracted (purple) and expanded (orange) orthogroups in the MRCA of each beetle tribe. Pictures correspond to the habitus of some surface-dwelling (left) and cave-dwelling (right) species included in the study: (I) *Limbodessus amabilis*, (II) *Limbodessus palmulaoides*, (III) *Paroster nigroadumbratus*, (IV) *Paroster macrosturstensis*, (V) *Catops fuliginosus*, (VI) *Leptodirus hochenwartii*. Photos from Watts and Hamon[31] (I and III), © Chris Watts and Howard Hamon (II and IV), © Udo Schmidt (V) and Hlaváč et al.[30]

(VI). Treemaps correspond to the enriched biological processes of the orthogroups that were contracted (orange), expanded (purple) or lost (pink) in the MRCA of each tribe. The size of the square is proportional to the *p* value obtained in the gene ontology (GO) enrichment analysis. Labels indicate manually summarized biological processes, see Supplementary Figs. 1, 2 and 3 for more details. **c** Venn diagrams indicating the percentage of OGs with a parallel evolution among the MRCAs of the three tribes and the total contracted, expanded, gained and lost OGs (see Supplementary Fig. 4a for the number of OGs of each overlap). **d** Treemap indicating the enriched functions of the orthogroups parallelly contracted in the ancestors of the three tribes (see Supplementary Fig. 4b for more detailed biological processes). Source data are provided as a Source Data file and at https://github.com/MetazoaPhylogenomicsLab/Balart_Garcia_et_al_2023_gene_repertoire_evolution_subterranean_coleoptera/blob/main/Source_data_2.zip.

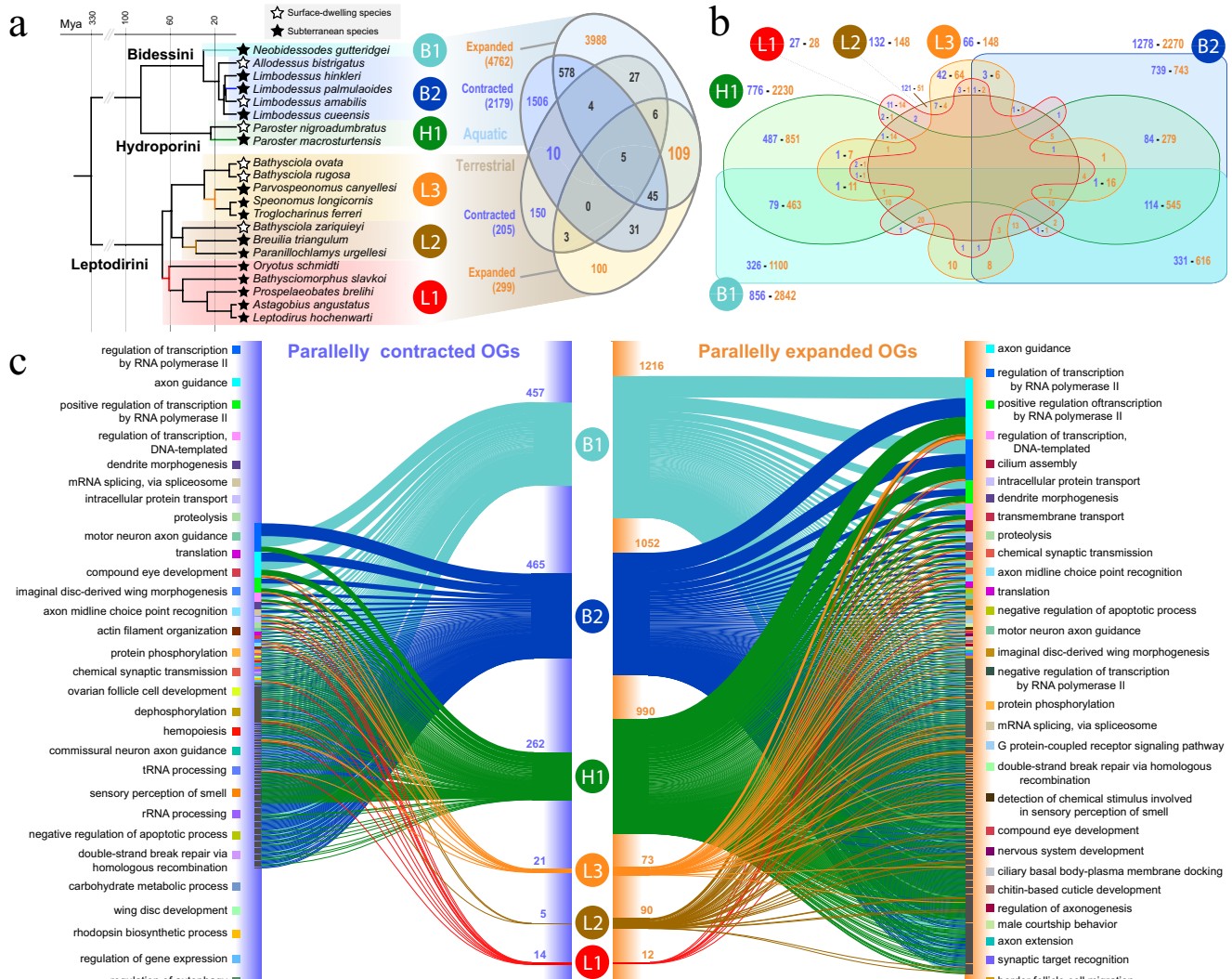

**Fig. 3 | Parallel evolution across subterranean Coleoptera. a** Phylogenetic context highlighting the six lineages that independently transitioned to underground habitats, named B1, B2 (Bidessini), H1 (Hydroporini), L1, L2 and L3 (Leptodirini). **b** Venn diagram indicating the number of contracted and expanded OGs (parallelly or not) in the six lineages (numbers indicated close to the labels of B1, B2, H1, L1, L2 and L3), and the number of OGs parallelly contracted (purple) and expanded (orange) between lineages (i.e., shared OGs by two or more lineages). **c** Alluvial plot showing the parallelly contracted OGs (purple) on the left side and the total parallelly expanded OGs (orange) on the right side for each of the lineages. Each line

represents an OG parallelly contracted/expanded and the bars on the sides correspond to the proportion of OGs with a similar function (sorted from the more represented functions at the top to the less represented functions at the bottom). Labels on the sides correspond to the top 30 more represented functions of the parallelly contracted/expanded OGs across subterranean beetle lineages. Source data are provided as a Source Data file and at https://github.com/MetazoaPhylogenomicsLab/Balart_Garcia_et_al_2023_gene_repertoire_evolution_subterranean_coleoptera/blob/main/Source_data_3.zip.

independent cave-dwelling lineages. Therefore, our results suggest the parallel loss of eyes and neural circuits associated to this complex structure in a large-scale subterranean evolutionary scenario may result from a genomic remodeling guided by both gene family contractions and expansions. Further developmental studies together with functional experiments would deepen our understanding about the nervous system evolution and sensory system rearrangements occurring in cave-dwelling beetles.

To test for functional convergent evolution between lineages, we explored and compared the annotations of the exclusively contracted and expanded OGs in each independent subterranean colonization. As shown in the Venn diagrams in Fig. 3a, b, the number of OGs with lineage-specific expansion and contraction was higher than that of OGs with parallel net rate shifts. In order to investigate if these lineage-specific contractions and expansions were functionally convergent, we inferred GO annotation similarity across all OGs with lineage-specific expansions or contractions, measured their functional similarity and grouped them into functional clusters that eventually represent functional convergence units (each functional cluster being represented as a polygon with a representative OG that acts as its centroid and that defines its main putative function; see "Methods"). Our results showed that most OGs with lineage-specific net rate shifts were grouped into functional clusters, indicating a high degree of functional convergent evolution between lineages (Fig. 4). The number of functional clusters in contracted OGs was three times lower than that of expanded ones, however the ratio of OGs per functional cluster was very similar (close to 10 OGs per functional cluster in both categories, Supplementary Data 2), indicating a similar degree of convergence both in expansions and contractions. Most of the exclusively expanded or contracted OGs underwent low to moderate changes in copy number (i.e., 99% of the expansions and contractions correspond to 1–5 gains and losses). At the functional level, convergently contracted or expanded OGs across lineages were related to sensory systems (e.g., perception of smell, compound eye development, rhodopsin biosynthetic process); metabolism (e.g., lipid and carbohydrate metabolic processes), nervous system development and functioning (e.g., axon guidance, axon extension, synaptic transmission), regulation of transcription and reproduction (e.g., sperm axoneme assembly, male courtship behavior), among others (Fig. 4a, b). These results thus point to convergent evolution mainly via gene family expansions as a key driver for underground specialization.

## The deep subterranean environment as a driver of parallel genomic innovation in terrestrial beetles

Despite our previous results indicating that the tribe Leptodirini has undergone no substantial changes in the gene repertoire evolutionary dynamics in the branches leading to the independent transitions to life in caves, but rather before these transitions, ancestral gene repertoire reconstruction indicated that large pulses of genomic change occurred in the branch leading to two Leptodirini subterranean lineages (Fig. 5a). These branches correspond to the split between highly specialized pairs of cave-dwelling species within Leptodirini, representing subterranean lineages that diverged from fully adapted ancestors which transitioned to the underground more than 30 Mya: (i) *Troglocharinus ferreri* - *Speonomus longicornis* and (ii) *Leptodirus hochenwartii* - *Astagobius angustatus*. These pairs of species correspond to sister genera that have independently developed a contracted life cycle (i.e., a single larval instar development instead of two or three instars as in the ancestral states)[36–38] and also show extreme anatomical modifications compared to their surface and less modified cave relatives, such as increased body size and elongation, and elongated sensory appendages (Fig. 5c). The changes in the gene repertoire dynamics were mainly driven by gene gain and duplication and largely corresponded to parallel changes in the same OGs, indicating parallel evolution as a driver of a higher degree of adaptation to life in caves.

Remarkably, the enriched GO terms associated with parallelly expanded OGs showed an enhancement of sensory perception, regulation of the nervous system processes, metabolic processes, locomotion, immune system processes, developmental processes, growth and reproduction among many other accentuated functions (Fig. 5b and Supplementary Fig. 5).

Together, gene copy-number variations and point mutations contribute to evolutionary novelty, through the process of gene duplication and divergence[39]. In order to assess how point mutations could potentially be contributing to parallel evolution in these highly adapted subterranean species of Leptodirini, we performed an exploratory analysis to detect positive selection in the expanded OGs in parallel (i.e., L1 and L3, Fig. 5a). For that, we leveraged the gene trees of each of these 216 OGs to test branches under positive selection in each subterranean lineage separately (see Methods). Our results indicate that 32.9% of the expanded OGs in parallel have a proportion of branches under positive selection in L1 and 23.1% in L3 (Supplementary Data 3). Furthermore, 12.5% of these OGs show parallel positive selection in both highly specialized cave-dwelling lineages, suggesting that a considerable proportion of parallel gene family expansions were further positively selected. Their functional annotation revealed putative roles in developmental and transcriptional regulation, neurogenesis, cuticle formation and signal transduction among other candidate functions (e.g., Homeobox, Cys2–His2 zinc finger, Cadherin, Pecanex C, ASH, Chitin Bind 4; Supplementary Data 3). Taken together, these findings support the idea of life in caves not as a 'wreck of ancient life', as coined by Darwin, but as an opportunity for genomic innovation through gene family expansion and point mutations, and provide an example of how gene duplication and divergence are key generators of evolutionary novelty linked to adaptation to subterranean life.

The genome-wide genetic underpinnings that facilitated adaptation to life in caves are still largely unknown for most cave fauna lineages. Here we show that genomic exaptation may have facilitated these transitions across beetles via parallel and convergent evolution of the gene repertoire prior to a transition underground. We found a contrasting pattern of evolution in aquatic and terrestrial cave beetles, with parallel evolution in the gene repertoires prevailing in aquatic subterranean lineages and convergent evolution rampant in terrestrial ones in the branches leading to transitions to subterranean environments. Our results show both gene gain and duplication accounted for a large part of the genomic changes prior to or leading to subterranean transitions. In particular, the significance of gene family expansions found in independent highly adapted subterranean lineages (and posterior accumulation of point mutations leading to positive selection) thus defies the widely accepted paradigm that conceives life in caves as evolutionary bottlenecks where loss of non-functional characters take place, with phenotypic regressions characterizing its inhabitants. In this study, we show that this interpretation of the phenotype should be better defined as a reshaping of the gene repertoire (and eventually of the phenotype) mainly driven by gene gain. Darwin's 'wrecks of ancient life' also rise, after all.

## Methods
### Data collection
No ethical approval was necessary for sample collection. Specimens were collected under permits 35601-58/2020-5 (Agencija RS za okolje, Ministrstvo za okolje in prostor, Republika Slovenija) and SF010263 (Department of Parks and Wildlife, WA). We collected and sequenced a total of 21 species, including 12 from the tribe Leptodirini, 1 from the tribe Catopini (i.e., a closely related tribe to Leptodirini), 2 from the tribe Hydroporini and 6 from the tribe Bidessini. The surface-dwelling species of the tribe Leptodirini and a species of the tribe Catopini (*Catops fuliginosus*) were collected in forest-litter using an entomological litter reducer and a metallic sieve of 3 mm aperture. The

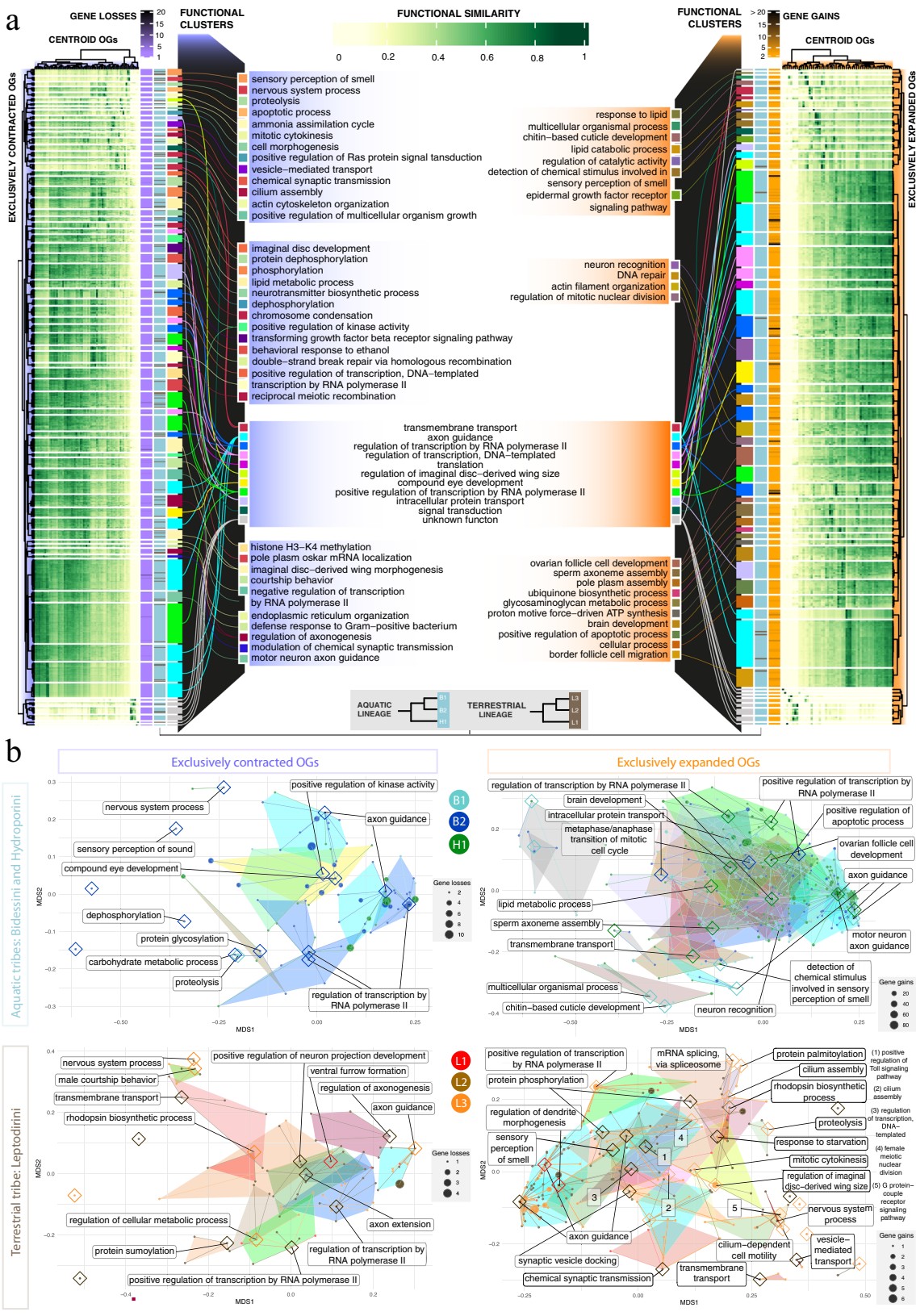

subterranean species were captured manually in caves using an entomological aspirator. The surface-dwelling species of the tribes Bidessini and Hydroporini were captured in shallow still water and temporary pools using aquatic nets. Groundwater-monitoring boreholes were used to access the groundwater habitat of the subterranean species of Bidessini and Hydroporini and specimens were captured

with a mesh net attached to a fishing rod. When the localities were remote, we kept specimens in RNA-later and cold conditions to reduce risk of RNA degradation. Otherwise, specimens were kept alive in thermo boxes with moss to keep the humidity (terrestrial beetles) or tubes with the same water where they were found (aquatic beetles). Samples were transported in dark and constant temperature

**Fig. 4 | Functional convergence between exclusively expanded and contracted OGs in each subterranean lineage. a** Heatmaps showing the functional similarity across exclusively contracted (purple) and expanded (orange) OGs in each of the six independent subterranean lineages (i.e., B1, B2, H1, L1, L2 and L3). Each functional cluster is represented by a set of OGs with a centroid OG that defines the main putative function (central labels). Functional clusters with the same annotation are represented with the same fill color. Functional similarity is measured by pairwise comparisons of all the gene ontology (GO) terms corresponding to biological processes. Side columns correspond to (i) total gene losses/gains, (ii) aquatic (light blue) or terrestrial (brown) lineage and (iii) the functional cluster. **b** Constellation plots representing the functional convergence between lineage-specific expanded and contracted OGs in two dimensions (see the functional similarity analysis in "Methods"). Dots represent the exclusively expanded or contracted OGs of each independent lineage and their size is scaled to the total number of gene gains and losses. Polygons correspond to the functional clusters obtained with the affinity propagation approach where the centroid OG is represented inside a diamond and its putative function is shown in the labels. Functional convergence is remarkable in processes related to nervous system, regulation of transcription, sensory systems/stimuli perception, metabolism and development. Source data are provided as a Source Data file and at https://github.com/MetazoaPhylogenomicsLab/Balart_Garcia_et_al_2023_gene_repertoire_evolution_subterranean_coleoptera/blob/main/Source_data_4.zip.

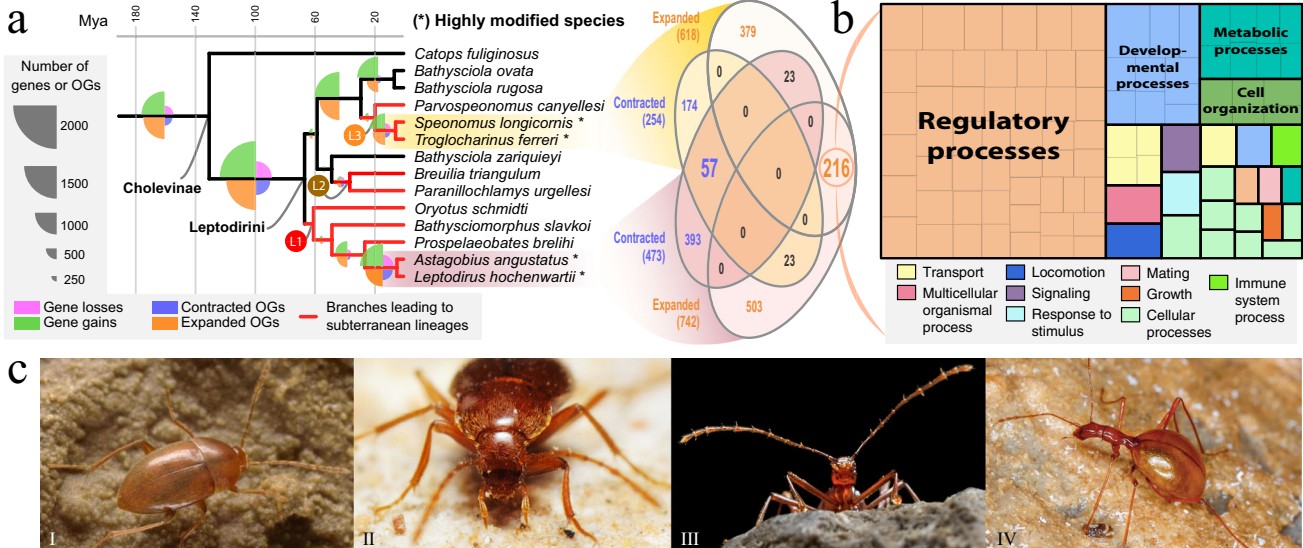

**Fig. 5 | Parallel evolution of the gene repertoires in two highly modified subterranean lineages of the tribe Leptodirini. a** Phylogenetic context highlighting the clades corresponding to the highly modified Leptodirini lineages (yellow and maroon colored ranges) and the represented species (*). The scaled pie charts in the branches indicate the total ancestral gene losses, gene gains, contracted and expanded orthogroups (OGs). **b** Treemap representing the overrepresented biological processes of the parallelly expanded OGs in the two independent highly modified lineages of Leptodirini obtained with a gene ontology enrichment analysis; the size of the squares is scaled to the *p*-value. **c** Pictures of the genera that represent the independent highly modified Leptodirini lineages: (I) *Speonomus longicornis*, (II) *Troglocharinus elongatus*, (III) *Astagobius angustatus* and (IV) *Leptodirus hochenwartii*. Photos: © Christian Vanderbergh (I), © Adrià Miralles-Nuñez (II), © Tin Rožman (III) and © Tvrtko Dražina (IV). Source data are provided as a Source Data file and at https://github.com/MetazoaPhylogenomicsLab/Balart_Garcia_et_al_2023_gene_repertoire_evolution_subterranean_coleoptera/blob/main/Source_data_5.zip.

simulating the conditions of their habitat. Specimens were identified and flash-frozen with liquid nitrogen. Samples were stored at −80 °C. Total RNA was extracted in cold, sterile and RNAse free conditions using different protocols, including (i) the selective Phenol/Chloroform extraction following a lysis through guanidinium thiocyanate buffer as described in Sambrook et al.[40] and modified in Balart-García et al.[20], (ii) the HigherPurity™ Tissue Total RNA Extraction Kit (Canvax Biotech) protocol and (iii) Qiagen Allprep RNA/DNA Isolation Kit. Total RNA yield was quantified with a fluorometer using the Qubit ® RNA HS Assay Kit (Thermo Fisher Scientific). For the species of the tribe Leptodirini, TruSeq stranded mRNA libraries were prepared. For the aquatic species, the single specimen samples were prepared with the Nugen kit. The multiple individual samples were prepared with the TruSeq kit at the Australian Genome Resource Facility (AGRF, Australia). A quality control check was performed using Agilent Technologies 2100 Bioanalyzer using the RNA Integrity Number (RIN). Libraries were sequenced in paired-end mode on the Illumina platforms HiSeq2500, and Novaseq 6000. *Limbodessus amabilis* data were obtained through an Iso-Seq protocol. A PacBio SMRTBell Template Prep Kit 1.0 SPv3, 1 "Non sized" selected library and 1 Size Selected library (4.5–10 kb) per sample was used. Libraries were sequenced in a PacBio Sequel II platform. mRNA sequencing was performed at the National Center of Genomic Analysis (CNAG, Spain), Macrogen (South Korea) and the Australian Genome Resource Facility (AGRF, Australia). For more detailed information about species, samples, localities, laboratory protocols, sequencing details see Supplementary Data 4.

We collected publicly available transcriptomic and genomic data of other Coleoptera species and two outgroups (i.e., Strepsiptera and Neuroptera) in order to include a broader phylogenetic context to study gene repertoire evolution. We also included an additional transcriptome of a cave-dwelling species of the tribe Leptodirini (*S. longicornis*) obtained from Balart-García et al.[20]. We selected the species based on the relationships of the main groups of Coleoptera obtained through recent phylogenomic approaches[41], data availability and the quality of the data based on the completeness score obtained after processing the raw data (see below). The raw data for each species was downloaded from public repositories and processed using the same pipeline as for the newly generated data (see next sections). For more detailed information about species and public data repositories see Supplementary Data 5.

**Data preprocessing**

We followed the raw data processing pipeline explained in Fernandez et al.[42] with some modifications. For RNA-seq data, adapters and low-quality reads were trimmed using fastp v.0.20.0[43], de novo assembly of clean data was carried out using Trinity v2.11[44] with default parameters,

candidate coding regions were predicted on assemblies, using Trans-Decoder v5.5.0 (https://github.com/TransDecoder/TransDecoder), contaminant sequences present in the transcriptome assemblies were filtered out using BlobTools2[45] (i.e., all non-metazoan sequences that were previously identified through a DIAMOND[46] BLASTP search (−sensitive −max-target-seqs 1 −evalue 1e-10) against the NCBI non-redundant (NCBI NR) protein sequence database (http://www.ncbi.nlm.nih.gov/)), and finally longest isoforms of each transcript were retained as final candidate coding regions for further analyses. For the PacBio data obtained for *L. amabilis* we used Iso-Seq v.3.1.2 (https://github.com/PacificBiosciences/IsoSeq) to process the raw reads, ANGEL (https://github.com/PacificBiosciences/ANGEL) to predict open reading frames, and CD-HIT v.4.8.1[47] to cluster highly identical sequences that represented isoforms with a similarity threshold of 0.9. For whole-genome sequencing data, we downloaded the predicted proteome for published genomes from the NCBI genome repository (https://www.ncbi.nlm.nih.gov/genome/). We assessed the quality and completeness of newly assembled transcriptomes and the down-loaded proteomes using BUSCO v4.1.4[48] in protein mode, obtaining an estimation of the completeness of the gene content based on the sets of benchmarking universal single-copy orthologs of the insecta data-base (insecta_odb10). The final set transcriptomes and genomes resulted in gene completeness score percentage higher than 80% (i.e., considering the sum of the percentages of both the complete and fragmented genes; mean 93.15% BUSCO completeness), with the exception of the newly generated transcriptomes of *Limbodessus cueensis* and *L. hinkleri* that resulted in a BUSCO completeness of 73% and 76%, respectively (see Supplementary Data 4 and 5 for further details). We acknowledge that lower BUSCO values may indicate a lower quality transcriptome, and in such cases downstream analysis could reflect either gene loss or context-dependent gene expression that was not captured in the samples. These two transcriptomes were included to provide more information for the ancestral reconstruction of the gene repertoire of Bidessini (see next sections). Nevertheless, when exploring parallel and convergent evolution during the inde-pendent subterranean transitions, we used the split between *L. amabilis* and *L. palmulaoides* as one of the two representative transi-tions of the tribe Bidessini (B2 in Fig. 3a), therefore minimizing any potential bias introduced by less complete gene sets.

## Orthology inference, phylogenetic analyses and molecular dating

We inferred orthology relationships between the proteomes ($n = 41$) using OrthoFinder v. 2.5.1[49], obtaining a total of 15,017 orthogroups (OGs hereafter) with more than 3 sequences. For each OG, we applied a pre-alignment quality screening to discard bad quality sequences using PREQUAL, which identifies and masks regions with non-homologous adjacent characters[50]. After this filtering, we discarded 29 OGs with less than 4 sequences and updated the table of sequence counts per OG obtained from the comparative genomics data of OrthoFinder (OG counts table hereafter; Supplementary Data 6). We aligned the amino acid sequences of the remaining 14,988 OGs using MAFFT v. 7.407[51] with a maximum of 1000 iterations. We inferred maximum-likelihood phylogenies for all these OGs that contained from 4 to 2211 sequences. To expedite computing, we divided the phylogenetic analyses in two approaches: (i) for the OGs with 400 sequences or more ($n = 36$), we inferred phylogenetic trees using FastTree v.2.1.1[52] with the LG + CAT substitution model; (ii) for the OGs with less than 400 sequences ($n = 14,952$), we used IQTREE v.1.6.12[53] using the ultrafast bootstrap approximation with 1000 replicates[54]. For the second approach, we firstly used ModelFinder[55] as implemented in IQTREE to find the best-fitting model for each OG based on the Bayesian Information Criterion (Supplementary Data 6). For the OGs that fitted into mixture models ($n = 10,515$), we generated a guide tree with FastTree (i.e., same version and model as explained above) that was

then used for the phylogenetic inference in IQTREE implementing the posterior mean site frequency (PMSF) model (i.e., site-specific fre-quency model)[56].

In order to obtain a topology for molecular dating, we generated a species tree using the following approach. We selected the OGs in which all the species of our data set were represented based on the OG counts table. The gene trees of the selected OGs ($n = 824$ OGs) and their corresponding alignments were processed with PhyloPyPruner v.1.2.4 (https://pypi.org/project/phylopypruner/), a software that fil-ters short sequences and removes paralogs using the species overlap method, obtaining pruned trees with 1:1 orthologs. We selected a minimum length of 100 amino acids and a minimum of 75% of species representation obtaining a total of 232 OGs. We used these 232 OGs to infer the species tree topology using the same phylogenetic approach explained in the previous section. Briefly, we used MAFFT to align the concatenated matrix, we used ModelFinder implemented in IQTREE to estimate the best-fitting model (i.e., LG + C60 + F + G), we used Fas-tTree to infer the guide tree and IQTREE to infer the final phylogenetic tree (i.e., same software versions and parameters as previously men-tioned). All nodes had more than 90% of bootstrap support except the split between *Onthophagus taurus* and *Hydrochus megaphallus* (55%) (Supplementary Fig. 6a). We followed the topology of McKenna et al.[41] (i.e., using a phylogenomic approach with a broader taxa representa-tion) for the placement of these taxa. Furthermore, to optimize the Bayesian estimation of species divergence times (i.e., molecular dating analysis hereafter), we subsampled the top 50 most informative OGs using GeneSortR, a software that uses a multivariate method to quantify phylogenetic usefulness[57]. The resulting 50 gene trees with-out paralogs, the final species tree topology and seven fossil calibra-tion points obtained from previous studies[38,58] (Supplementary Table 1) were used to infer a time-calibrated tree with MCMCtree[59], using the approximate likelihood method[60] and implementing two alternative runs under two different molecular clocks: an independent rate clock and an autocorrelated rate clock (Supplementary Fig. 6b). Finally, we revised the congruence of the results with each molecular clock with Tracer v.1.7.1[61]. We used iTOL v6 to visualize and render the phylogenetic trees[62].

## Estimation of gene gain, duplication and loss and reconstruction of the ancestral gene repertoire of the MRCA of Leptodirini, Hydroporini and Bidessini

We reconstructed gene gains and losses per branch per million years using BadiRate v.1.35 under the birth, death and innovation model (BDI). This maximum-likelihood-based approach estimates gene gains and losses (net rates hereafter) for a set of gene families in a given phylogenetic context[63]. We used the OG counts table (i.e., obtained with OrthoFinder and updated after the PREQUAL filtering) and the time-calibrated tree based on the independent rate clock run as an input for the BadiRate analysis. Since BadiRate will crash with highly heterogeneous gene counts, we ran BadiRate for each OG separately (i.e., only 0.01% failed). Moreover, we computed and compared the results of two different implementations in order to detect statistically significant changes in the net rates: (i) a global rates model, that assumes that all branches evolve under the same net rate and (ii) an branch-specific rates model, which assumes that a given set of bran-ches have a different rate than the background (in our case branches leading to subterranean lineages). Furthermore, we performed the analysis at three phylogenetic levels separately as follows: (i) including all taxa (i.e., 14,980 OGs), (ii) including just the Dytiscidae species (i.e., 8993 OGs), and (iii) including just the Cholevinae species (i.e., 10,217 OGs). We extracted the best-fitting model by calculating the Akaike Information Criterion (AIC) from the resulting likelihood values and comparing the AIC under each model. We then obtained a list of sta-tistically significant OGs under the branch-specific rates model, therefore indicating significant changes in the net rates of

subterranean lineages compared to the background. In addition, we extracted the 'birth' and 'death' rates (i.e., indicative of gene family expansion or contraction, respectively) and calculated the net rate between the global rates model and the branch-specific rates model, thus obtaining an estimate of the magnitude of the expansion or contraction of each OG in the subterranean lineages compared to the background (Supplementary Data 7).

Furthermore, we parsed the number of gene gains and losses at each branch for each OG based on the results obtained under the best-fitting model (i.e., global rates or branch-specific rates). We calculated the total of contracted and expanded OGs for each branch with a custom R script (get_badirate_gains_losses.R). We generated lists of contracted, expanded, gained and lost OGs at each branch of interest (i.e., MRCA of each beetle tribe, the branches representing the habitat shift and the branches of the highly modified terrestrial beetle lineages, as defined in the main text). To generate lists of lost OGs, we compared the OGs present in a certain branch with the OGs present in the previous branch to identify and parse the missing ones (e.g., the lost OGs of the MRCA of Leptodirini corresponding to those present in the MRCA of Cholevinae but not present in the consecutive branch) (see Supplementary Data 7). We used these lists to explore parallel evolution (i.e., the same OGs with the same evolutionary dynamic changes between different lineages) and convergent evolution (i.e., different OGs exclusive to each lineage but with similar evolutionary dynamics and function).

## Functional annotation, gene ontology enrichment and functional similarity analysis

We annotated the amino acid sequences of each proteome using eggNOG-mapper v.2.1.6[64], which uses a database based on pre-computed orthologous groups and phylogenies to transfer functional information from fine-grained orthologs only. We used the DIAMOND algorithm for the search step using the Insecta database. We extracted gene ontology (GO) terms of each sequence for each OG and filtered unique GO terms per OG in order to explore their candidate functions. To reduce the complexity of the GO information, we only used the biological process category to represent graphically and explore the most relevant functions in the OGs of interest (see Supplementary Data 8 for annotations at the level of cellular component or molecular function).

To explore the global overrepresented functions at each evolutionary timepoint or transition, (i.e., in the MRCA of the beetle tribes or in the highly modified lineages of the tribe Leptodirini), we performed GO enrichment analyses with GOEnrichment v.2.0 (https://github.com/DanFaria/GOEnrichment) with the go-basic.obo database (http://geneontology.org/), and two-tailed Fisher's tests using the Benjamini-Hochberg correction method to discard false discovery rate (FDR)[65] (q-value > 0.01). We used the branch-specific OG lists (i.e., contracted, expanded, lost or gained) as a query and branch-specific backgrounds based on the existing OGs at each branch that contained GO annotations. These OG lists and backgrounds were obtained based on the BadiRate reconstruction explained in the previous section. For instance, to perform a GO enrichment analysis on the expanded OGs in the MRCA of the tribe Leptodirini we used the list of expanded OGs in that branch and all the OGs that contained GO term information as a background. For visualizing and summarizing the GO enrichment results, we used REVIGO web interface[66], using the *Drosophila melanogaster* database and the SimRel method to estimate the semantic similarity across enriched GO terms.

In order to explore functional convergence in the lineages transitioning to subterranean habitats (i.e., B1, B2, H1, L1, L2 and L3; Fig. 4), we followed a similar approach to that described in Aristide and Fernández[67] and used the R package constellatoR (https://github.com/MetazoaPhylogenomicsLab/constellatoR). In brief, we first obtained lists of exclusively expanded and contracted OGs in each lineage and we then measured their functional similarity based on pairwise semantic similarity measures of all the GO terms in each OG, following the GOSemSim v.3.16 approach[68]. We obtained two pairwise OG similarity matrices containing (i) all the exclusively expanded OGs measures, and (ii) all the exclusively contracted OGs measures separately (Supplementary Data 1). In order to detect functional clusters, we first transformed our similarity matrices into distance matrices and applied a classic multidimensional scaling approach based on euclidean distances. Then we used the R package apcluster v.1.4.10 (https://cran.r-project.org/web/packages/apcluster) to obtain functional clusters through the affinity propagation method[69], an exemplar-based agglomerative approach that takes measures of similarity between pairs of data points and calculates a set of optimal exemplars and clusters gradually. In the context of this study, a functional cluster is thus defined as a set of OGs that converge with a similar function and contain an exemplar OG that acts as a cluster centroid and defines the main putative function (referred to as centroid OGs in Fig. 4). To visualize the functional convergence results including all lineages together we used the R package ComplexHeatmap v.3.15[70] (Fig. 4a). To visualize the lineage-specific results we used the R package ggplot2[71], using scatter plots where the functional clusters are represented as polygons (i.e., a graphical representation that we called 'constellation plots'; Fig. 4b). We restricted the list of all exclusively expanded OGs (n = 2000) to those with more than one gene gain (n = 370) (Fig. 4a), and we did the same for the exclusively expanded and contracted OGs in the aquatic lineage (Fig. 4b), for reasons of feasibility in terms of graphical display (see Supplementary Data 1 and 2 for the results with the complete lists). To assign a representative function for each functional cluster, we used the most specific GO term (i.e., biological process) of the exemplar OG in a given cluster (i.e., reduced GO term hereafter). To obtain the reduced GO term for each OG, we performed a two steps approach combining the R package GOexploreR v.1.2.6[72] to navigate the GO hierarchy across all the GO terms of a given OG and extract the three terms at the lower level (i.e., the more specific biological processes), and the R package rrvgo v.1.8.1 (https://ssayols.github.io/rrvgo) with a threshold of 0.95 of semantic similarity obtaining a single GO term per OG. These reduced GO terms were calculated for all the OGs and were also used to explore the candidate functions of the parallelly expanded and contracted OGs (Fig. 3).

## Positive selection analysis

We tested branches under positive selection in the 216 expanded OGs in parallel across the highly modified subterranean lineages of the tribe Leptodirini (Fig. 5a). Firstly, we pruned the previously inferred gene trees of each OG obtaining subtrees for L1 and L3 branches separately using the R package ape. To obtain codon-aware alignments, we first aligned the amino acid sequences with MAFFT v. 7.407[51] and then used these alignments as a scaffold to guide the alignment of the nucleotide sequences in HyPhy[73]. Secondly, we used the exploratory option of the aBSREL method[74] to detect branches under positive selection in each phylogeny based on the dN/dS metric. P-values at each branch were corrected for multiple testing using the Holm-Bonferroni correction. Finally we extracted the proportion of branches under positive selection and calculated the percentage with the total tested branches per OG for each lineage (i.e., L1 and L3), and checked their candidate functions based on the eggNOG-mapper annotations (Supplementary Data 3). We extracted the reduced GO term for each OG as previously described and also included the best annotation hits of each OG based on the p-values obtained with eggNOG-mapper for all the sequences of the species represented in the highly modified lineages of Leptodirini.

## Reporting summary

Further information on research design is available in the Nature Portfolio Reporting Summary linked to this article.

## Data availability

The raw data generated in this study have been deposited in the National Center for Biotechnology Information database under Bio-Project accession code PRJNA902350. Accession numbers for the publicly available data analyzed in this study are indicated in Supplementary Data File 4. Remaining tissue samples were deposited at the cryocollection of the Metazoa Phylogenomics Lab at the Institute of Evolutionary Biology (CSIC-UPF). Source data are provided with this paper.

## Code availability

All processed data and custom scripts have been deposited in github: https://github.com/MetazoaPhylogenomicsLab/Balart_Garcia_et_al_2023_gene_repertoire_evolution_subterranean_coleoptera/releases/tag/v1, https://doi.org/10.5281/zenodo.8026413

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

## Acknowledgements

P.B.G. was supported by an FPI grant (grant agreement no. BES-2017-081050) financed by MCIN/AEI /10.13039/501100011033 and by European Social Fund (ESF) 'Investing in your Future' associated to the project CGL2016-76705-P funded by MCIN/ AEI/10.13039/501100011033/ and by ERDF 'A way of making Europe'. PBG also acknowledges support from the Systematics Research Fund 2020 (Linnean Society of London and the Systematics Association). R.F. acknowledges support from the following sources of funding: Ramón y Cajal fellowship (grant agreement no. RYC-2017-22492 funded by MCIN/ AEI /10.13039/501100011033 and ESF 'Investing in your future'), project PID2019-108824GA-I00 funded by MCIN/AEI/10.13039/501100011033, and by the European Research Council (ERC) under the European's Union's Horizon 2020 research and innovation program (grant agreement no. 948281). S.J.B.C. acknowledges support from the Australian Research Council (grant agreement DP18010385 and DP120102132). We thank Bill Humphreys and Chris Watts for collections of dytiscid species and Jordi Comas, Javier Fresneda and David Sánchez for their support in sampling leiodid species. We also thank Centro de Supercomputación de Galicia (CESGA) for access to computer resources. We acknowledge Ignacio Ribera for his invaluable contribution at the beginning of this project, making this research possible by conceiving the study, providing resources and supervising this research until the end of his life.

## Author contributions

P.B.G. and R.F. designed and conceived the research. P.B.G., S.P. and S.J.B.C. collected samples and provided valuable knowledge about the studied systems. T.M.B. and P.G.B.H. generated genomic data of the aquatic lineages. PBG generated genomic data of the terrestrial lineages, processed both newly generated and public genomic data and performed all analyses. L.A. assisted in the phylogenomic analyses and

graphical representation of the functional convergence analysis. P.B.G. prepared the figures and other provided materials. P.B.G., S.J.B.C. and R.F. interpreted and discussed the results. S.J.B.C. and R.F. provided resources. R.F. supervised the study. P.B.G. and R.F. drafted the paper. All authors contributed to the final version of the manuscript.

## Competing interests

The authors declare no competing interests.
