## [Peer Review File · Nature Communications]

Parallel and convergent genomic changes underlie independent subterranean colonization across beetlesReviewers' Comments:

Reviewer #1:

Remarks to the Author:

Review of Genomic exaptation and convergent evolution paved the way to independent subterranean colonization across beetle lineages by Balart-García and colleagues:

During the past 280 million years, beetles have evolved incomparable biodiversity, morphological disparity, and ecological diversity. Convergence is one of the driving forces that boost the radiation of beetles, enabling beetles to occupy nearly all niches we can imagine on Earth.

Cave-dwelling in beetles has apparently evolved multiple times independently as a result of key selective environmental constraints inherent to the subterranean lifestyle such as the absence of light (loss or reduction of eyes and pigmentation). As such, cave-dwelling beetles are one of the best systems to explore the genomic basis of convergent evolution associated with a strictly subterranean lifestyle. Genomic basis underlying the evolution of cave-related morphologies are poorly studied in the meta-genomic era. Balart-García and coauthors sequenced, investigated and characterized evolutionary dynamics of gene repertoire evolutionary dynamics across surface-dwelling and subterranean species that represent multiple independent underground colonizations both in terrestrial (Leptodirini) and aquatic beetle lineages (Hydroporini and Bidessini), representing three tribes of two remotely related beetle families. The results clearly showed that significant genomic changes largely driven by gene family expansions occurred prior to underground colonization in these beetles, indicating that genomic exaptation is responsible for a strict subterranean lifestyle parallelly across the studied beetle lineages. This study is a textbook example showing the genome underpinnings of parallel and convergent changes in the evolutionary dynamics of the gene repertoires of multiple insect lineages. These findings shed new lights onto our understanding the evolution of beetle biodiversity and adaption of life to environment from a meta-genomic perspective.

1. As the gene family expansions is the key factor in the evolution of independent highly adapted subterranean lineages, would it possible to compare your results with other lineages of insects such as ants (Hymenoptera)? Romiguier et al. (2022) sequenced 65 genomes to produce a phylogenetic tree of all ant subfamilies, with the subterranean leptanillomorph clade as the sister group of all other ants. Is similar pattern of gene family expansions found in both subterranean ants and beetles? Can we abstract a more general conclusion regarding the genomic adaptation of insects to subterranean environments? <https://doi.org/10.1016/j.cub.2022.05.001>

2. Some of the sequenced beetles, *Limbodessus* spp., have a comparative low BUSCO completeness that seem to be not ideally for downstream analyses. How did this affect the prediction of gene gain, duplication and loss?

Minor points:

1. 3mm should 3 mm
2. Australian AGRF, what is AGRF short for?
3. In many places, hyphen should be replaced by en dash.

Reviewer #2:

Remarks to the Author:

By contributing 22 new genome sequence assemblies for a large number of highly cave-adapted beetles species, the work described in the manuscript by Balart-García et al. constitutes a transformational contribution to the field of cave biology. As much as I can judge, data generation, documentation, and presentation meet state-of-the-art standards. There are, however, some major and minor issues to address for the manuscript to reach its full potential for the wide readership it deserves.

Major issues:

1.

"Here, we explore the genomic underpinnings of adaptations to life in caves in Coleoptera using a genome-wide phylogenomic approach."

For understandable reasons, the authors focused their analysis on gene family evolution dynamics. The coding partitions of animal genomes, however, rarely exceed 2% of eukaryote genomes. Animal genomes are occupied by immense partitions of mobile DNA and satellite DNA. As presented, the study presents a phylogenomic comparative framework for the comparative analysis of gene family evolution but falls short of a genomic study in a strict sense.

Some basic stats on the non-coding partitions is essential bc population genetic models and previously published studies suggest that a reduction in effective population size results in the increase of genome sizes due to the higher impact of genetic drift on the fixation of slightly deleterious sequence change, such as mobile DNA expansion (Lynch et al., 2011; Lynch & Conery, 2003). A genomic analysis of cave adaptation is therefore expected to examine the expectation that the genomes of highly endemic troglobites are larger and more mobile DNA enriched compared to closely related surface species. The same is true for satellite DNA.

These analyses will also speak to the authors' interest in gene family evolution bc extreme population bottlenecks at the dawn of cave colonization may be an explanation for the exceptional gene family size changes they detected in the cave lineage MRCAs.

Another crucial genome partition to look at are pseudogenes as the most recent direct evidence of processes in the process of regressive evolution. This data source is one of the most direct ones regarding the relative contributions of regressive vs constructive trait change in cave adaptation, which is a major question in the field at this point.

Recent work by the Cooper lab nicely pinpointed vision-related pseudogenes in the subterranean diving beetle genera included in this submission (Langille et al., 2022). From the perspective of sequence quality assessment as well as evolutionary informativeness, it seems an obvious question whether these were captured in the new genome sequence drafts.

In extension, it should be noted that even actual gene loss events could be pinpointed and quantified with high confidence through the comparative analysis of synteny traces.

In light of these gaps, I think the "mainly" in the statement "Our results indicate that remarkable genomic changes mainly driven by gene family expansions ..." is not sufficiently supported at this point. Phenotypic evolution results from both cis-regulatory changes and protein sequence changes besides gene family size changes (Stern, 2011). At this point, we study does not provide insights into the relative contribution of cis-regulatory DNA change to cave adaptation in their lineages.

2.

"Our results thus indicate that the MRCAs of the three tribes experienced a significant level of gene gain and duplication, and suggest that genomic exaptation fueled by an expansion of the gene repertoire may have paved the way for a later colonization of the underground in these three tribes."

The authors adequately define their gene exaptation model as a possibility at this point. I would agree that it is not yet possible to conclude with high confidence that "exaptation" of duplicated genes for cave traits has taken place. For this, we would need to know about their functionality as nascent gene duplicates and how this ancestral function compares to that in cave-adapted descendants. The authors should consider making attempts in this direction by looking at some well-understood genes in their

data set.

While I consider the analysis of gene family size change histories as thorough and highly reliable, it is important to remember that gene duplication outcomes can be phenotypically neutral (Lynch & Conery, 2000). In extension, GO-term enrichment in expanded gene family populations can result from both adaptive effects or tolerance of gene dosage increase. Selection tests are required to distinguish these very different scenarios. This can be as straightforward as relative rate tests or dN/dS. I think this is a compulsory additional analysis the authors need to carry out bc it has a high potential to clarify the adaptive significance of the duplicated gene partition. The opportunity to explore paralog diversification is a present that comes along with gene duplications.

Also note that while there is rationale to consider exaptation and cooption equivalent terms, exaptation in a strict sense does not speak to whether a new gene function completely replaced an ancestral one, while cooption unambiguously implies the conservation of ancestral functionality during the acquisition of new additional functions. Both scenarios are conceivable. My understanding of the literature is that cooption is the far more frequently documented trajectory. So the authors should include this distinction and ideally test for them for a few genes with well-documented functions.

3.

"Adaptation to life in caves is often accompanied by dramatically convergent changes across distantly related taxa, epitomized by the loss or reduction of eyes and pigmentation. Nevertheless, the genomic underpinnings underlying the evolution of cave-related phenotypes are largely unexplored."

Following up on this statement, the authors do cite previous work that reported pivotal genetic changes underlying cave adaptive trait changes. However, I do feel this presentation falls short of giving adequate oversight on the essential insights gained through the genetic and genome-wide approaches in cavefish. It will be necessary to fill this gap to be able to illustrate the general significance of their new work in the discussion section.

4.

"This parallel pulse of gene gain and loss in independent highly specialized subterranean lineages supports the idea of life in caves not as a 'wreck of ancient life', as defined by Darwin, but actually as an opportunity for genomic innovation through gene family expansion."

+

"we show that this interpretation of the phenotype should be better defined as a reshaping of the gene repertoire (and eventually of the phenotype) mainly driven by gene gain. Darwin's 'wrecks of ancient life' also rise, after all."

Given the lack of functional insights at this point, I don't think it is clear how much the investigated cave beetle lineages are truly genetically more complex compared to the MRCA's with their surface relatives. Independent of the answer, it would not change the previously characterized ratio of regressive vs constructive trait changes at the level of the phenotype.

Finally, as mentioned above, the question of how much adaptive vs non-adaptive processes are part of cave-adaptive transitions remains one of the core questions in the field. The authors understandably and convincingly pinpoint the parallel expansions of gene repertoires in their study. They also emphasize the important finding that these expansions preceded the actual process of cave colonization. In my mind, an intriguing non-adaptive explanation for these findings is the occurrence of extreme population size shrinkages of the source lineages that managed to persist before or in the process of retreating into cave ecologies. However, other scenarios are conceivable. Only further

insights into the adaptive vs non-adaptive outcomes of the detected gene duplications have the high potential to provide functional as opposed to corollary insights.

Minor issues:

1.

"Furthermore, gene gains and duplications were more abundant..."

Does the listing of "gene gain" next to "duplication" refer to gene duplication-independent processes?

2.

"genome-wide phylogenomic approach"

This seems redundant to me. Phylogenomic implies genomic

3.

The category "compound eye development" is captured in GO terms that both contracted and expanded in parallel in Fig. 3c. It would be helpful if these genes are explored for an explanation of this opposite trends.

4.

"In the context of this study, we refer to parallel evolution as significant shifts in net rates occurring in gene families shared by all or several lineages. In contrast, we define convergent evolution as lineage-specific significant changes in net rates in gene families that have similar putative functions when compared across lineages."

(Scotland, 2010) should be cited for a clear definition of parallel vs convergent which matches that of the authors.

Langille, B. L., Tierney, S. M., Bertozzi, T., Beasley-Hall, P. G., Bradford, T. M., Fagan-Jeffries, E. P., Hyde, J., Leijs, R., Richardson, M., Saint, K. M., Stringer, D. N., Villastrigo, A., Humphreys, W. F., Austin, A. D., & Cooper, S. J. B. (2022). Parallel decay of vision genes in subterranean water beetles. *Molecular Phylogenetics and Evolution*, 173, 107522.

Lynch, M., Bobay, L.-M., Catania, F., Gout, J.-F., & Rho, M. (2011). The repatterning of eukaryotic genomes by random genetic drift. *Annual Review of Genomics and Human Genetics*, 12, 347–366.

Lynch, M., & Conery, J. S. (2000). The evolutionary fate and consequences of duplicate genes [Review of The evolutionary fate and consequences of duplicate genes]. *Science*, 290(5494), 1151–1155. sciencemag.org.

Lynch, M., & Conery, J. S. (2003). The origins of genome complexity. *Science*, 302(5649), 1401–1404.

Scotland, R. W. (2010). Deep homology: a view from systematics. *BioEssays: News and Reviews in Molecular, Cellular and Developmental Biology*, 32(5), 438–449.

Stern, D. L. (2011). *Evolution, Development, and the Predictable Genome*. W. H. Freeman.

Reviewer #3:

Remarks to the Author:

Overall, this paper is well-written and interesting. It is likely to have a notable and positive impact on insect/beetle comparative genomics, including our understanding of the evolution of cave-dwelling habits. While there have been many studies of cave-dwelling insects, including beetles, none to my knowledge have undertaken a comparative genomic study involving multiple distantly-related species with convergent habits. The choice of taxa is interesting because it spans the basal split in Coleoptera and the reconstructed phylogeny for Coleoptera is consistent with the best available estimates from other studies. Most of the authors conclusions are based on homology to genes of known function in distantly-related species and many conclusions are generalizations, leaving much uncertainty. However, I cannot conceive of a suitable remedy for this at the taxonomic scale at which they are working and I consider the results both interesting and important. Their results suggest many intriguing paths for future study, including functional validation/characterization of genes proposed to be involved in specializations for cave life, notably including metabolism, vision, and development. The methodology is sound and meets the expected standards in phylogenetic systematics and comparative genomic analyses. I have only a few suggestions (see below) for improvement of the text, which is generally very well-written. Note, line numbering would have been helpful.

Introduction:

The first sentence appears to be an incomplete thought. Rewrite for clarity.

Page 2: The following sentence is unclear/incomplete: Diving beetles (Dytiscidae) also represent a large radiation in subterranean aquatic ecosystems; they colonized independently multiple underground desert aquifers in Western Australia being this colonization 'recent' (ca. 12 - 9 Mya respectively).

Page 2:

It would be helpful if you indicated here the beetle families to which these tribes belong, as well as their estimated divergence times from other taxa studied.

Results & Discussion:

Line 1: The statement "the main beetle lineages plus two outgroups" is too vague. What lineages were sampled (suborders, series or superfamilies; were any series or superfamilies missing?) and what outgroup taxa were sampled? This is elsewhere in your paper/supplement, but is not easy/quick to find nor apparent in the labeling of your phylogeny.

Need commas after "i.e." and "e.g." throughout.

Figs 4a 4b too cluttered and complicated. Informative, but overwhelming to look at. I cannot immediately recommend a good solution to this, but I hope you will consider rethinking how these data are presented to both make it easier for the reader to interpret and to avoid overwhelming the reader with too much information. At a minimum, I recommend directing the reader (in the legend) to your most notable results.

Methods:

I would rethink citing Cai et al. (2022) for its topology. There are problems with the analyses in that paper and the subordinal relationships it recovered differ from those in your paper as well as other recent phylgenomic studies. See Boudinot et al. (in press Syst. Ent.; <https://doi.org/10.1111/syen.12570>) for more information. Nonetheless, I think it is reasonable to cite Cai et al. 2022 for fossils.

REVIEWER COMMENTS - AUTHORS RESPONSE

Reviewer #1 (Remarks to the Author):

Review of Genomic exaptation and convergent evolution paved the way to independent subterranean colonization across beetle lineages by Balart-García and colleagues:

During the past 280 million years, beetles have evolved incomparable biodiversity, morphological disparity, and ecological diversity. Convergence is one of the driving forces that boost the radiation of beetles, enabling beetles to occupy nearly all niches we can imagine on Earth.

Cave-dwelling in beetles has apparently evolved multiple times independently as a result of key selective environmental constraints inherent to the subterranean lifestyle such as the absence of light (loss or reduction of eyes and pigmentation). As such, cave-dwelling beetles are one of the best systems to explore the genomic basis of convergent evolution associated with a strictly subterranean lifestyle. Genomic basis underlying the evolution of cave-related morphologies are poorly studied in the meta-genomic era. Balart-García and coauthors sequenced, investigated and characterized evolutionary dynamics of gene repertoire evolutionary dynamics across surface-dwelling and subterranean species that represent multiple independent underground colonizations both in terrestrial (Leptodirini) and aquatic beetle lineages (Hydroporini and Bidessini), representing three tribes of two remotely related beetle families. The results clearly showed that significant genomic changes largely driven by gene family expansions occurred prior to underground colonization in these beetles, indicating that genomic exaptation is responsible for a strict subterranean lifestyle parallelly across the studied beetle lineages. This study is a textbook example showing the genome underpinnings of parallel and convergent changes in the evolutionary dynamics of the gene repertoires of multiple insect lineages. These findings shed new lights onto our understanding the evolution of beetle biodiversity and adaptation of life to environment from a meta-genomic perspective.

> Authors: We thank the reviewer for their comments and are glad they found our results interesting.

1. As the gene family expansions is the key factor in the evolution of independent highly adapted subterranean lineages, would it possible to compare your results with other lineages of insects such as ants (Hymenoptera)? Romiguier et al. (2022) sequenced 65 genomes to produce a phylogenetic tree of all ant subfamilies, with the subterranean leptanillomorph clade as the sister group of all other ants. Is similar pattern of gene family expansions found in both subterranean ants and beetles? Can we abstract a more general conclusion regarding the genomic adaptation of insects to subterranean environments? <https://doi.org/10.1016/j.cub.2022.05.001>

> Authors: We agree with the reviewer that it would be interesting to compare the patterns of gene family expansion we found in subterranean beetles with other insects with a similar ecology. Rominguier et al. (2022) explored the genetic innovations associated with the origin of eusociality and focused on describing the genomic changes occurring in the branch leading to formicoids. Moreover, their phylogenetic reconstruction suggests a potential subterranean origin for ants. Unfortunately, they do not describe the gene repertoire evolutionary dynamics in the leptanillomorph clade (subterranean) and in its most recent

common ancestors, thus the comparison with the findings of the present study cannot be directly addressed. They added the results to the supplementary material and we think it would be interesting analyses to carry out. However, we do not consider such an option suitable in this case as it would require re-processing and analyzing of the relevant data and a complete rewrite of our manuscript. While gene family evolution, particularly family expansions, have been found to be related to the adaptation of insects to multiple ecological niches (e.g., Freitas and Nery, 2020; Kim et al. 2022), they have not yet been studied in the context of adaptation to the subterranean lifestyle. For example, and likewise to what was found by Rominguier et al. (2022), gene duplication has been proposed to be relevant in the social evolution of termites (Shigenobu, 2022). Moreover, socially parasitic ant species show contraction in gene families involved in chemoreception, cuticular formation, and composition among others (Schrader et al. 2021). As our study represents the first look at the evolution of the whole coding-gene repertoire in a group of cave insects, we can only compare our results to those found in distantly related groups of animals inhabiting subterranean environments, such as cavefish or crustaceans, which belong to different evolutionary scenarios and have fundamentally different biologies, and therefore we consider them not fully appropriate for this study.

2. Some of the sequenced beetles, *Limbodessus* spp., have a comparative low BUSCO completeness that seem to be not ideally for downstream analyses. How did this affect the prediction of gene gain, duplication and loss?

> Authors: As the reviewer correctly points out, the precise characterization of the evolution of the gene repertoire can be affected by errors stemming from various sources in the identification of each species gene set. We indeed used BUSCO completeness scores as a measure, admittedly imperfect, of this error and, in some cases, we decided to discard some of the newly generated transcriptomes due to low completeness scores. Nonetheless, applying a phylogeny-aware method for the reconstruction of gene duplications, gains and losses confers robustness to our gene repertoire reconstructions, as these methods are in general robust to species-specific deviations (e.g. due to measurement error), considering instead the whole evolutionary context for inference (Felsenstein 1985). In this sense, our approach is conservative since we search for signals in the branches leading to subterranean lineages, and also precisely what we see is a burst of gene gain and duplication, which could not stem from species-specific artifacts in gene repertoire due to incomplete transcriptomes. Moreover, when exploring parallel and convergent changes during the independent subterranean transitions, we used the split between *L. amabilis* and *L. palmulaoides* as representative of a subterranean transition in Bidessini (B2 in Fig. 3a). The transcriptomes of these species had relatively higher completeness scores and allowed us to avoid any potential bias introduced by *L. hinkleri* and *L. cueensis*. The inclusion of these two transcriptomes has an effect in the reconstruction of the MRCA of Bidessini by providing more information about the gene content of this lineage, but the reconstruction of B2 (transition to caves) is not affected. We have now added this clarification in the methods of the manuscript.

Minor points:

1. 3mm should 3 mm

2. Australian AGRF, what is AGRF short for
3. In many places, hyphen should be replaced by en dash.

> Authors: We have corrected these typos and added information about what the acronym means (Australian Genome Resource Facility, Page 13).

References:

- Freitas, L. & Nery, M. F. Expansions and contractions in gene families of independently-evolved blood-feeding insects. *BMC Evol. Biol.* **20**, 87 (2020).
- Kim, H. *et al.* Gene family expansions in Antarctic winged midge as a strategy for adaptation to cold environments. *Sci. Rep.* **12**, 18263 (2022).
- Shigenobu, S. *et al.* Genomic and transcriptomic analyses of the subterranean termite *Reticulitermes speratus*: Gene duplication facilitates social evolution. *Proc. Natl. Acad. Sci. U. S. A.* **119**, (2022).
- Schrader, L. *et al.* Relaxed selection underlies genome erosion in socially parasitic ant species. *Nat. Commun.* **12**, 2918 (2021).
- Felsenstein, J. Phylogenies and the comparative method. *Am. Nat.* (1985).

Reviewer #2 (Remarks to the Author):

By contributing 22 new genome sequence assemblies for a large number of highly cave-adapted beetles species, the work described in the manuscript by Balart-García *et al.* constitutes a transformational contribution to the field of cave biology. As much as I can judge, data generation, documentation, and presentation meet state-of-the-art standards. There are, however, some major and minor issues to address for the manuscript to reach its full potential for the wide readership it deserves.

Major issues:

1.

“Here, we explore the genomic underpinnings of adaptations to life in caves in Coleoptera using a genome-wide phylogenomic approach.”

For understandable reasons, the authors focused their analysis on gene family evolution dynamics. The coding partitions of animal genomes, however, rarely exceed 2% of eukaryote genomes. Animal genomes are occupied by immense partitions of mobile DNA and satellite DNA. As presented, the study presents a phylogenomic comparative framework for the comparative analysis of gene family evolution but falls short of a genomic study in a strict sense.

Some basic stats on the non-coding partitions is essential bc population genetic models and previously published studies suggest that a reduction in effective population size results in the increase of genome sizes due to the higher impact of genetic drift on the fixation of slightly deleterious sequence change, such as mobile DNA expansion (Lynch *et al.*, 2011;

Lynch & Conery, 2003). A genomic analysis of cave adaptation is therefore expected to examine the expectation that the genomes of highly endemic troglobites are larger and more mobile DNA enriched compared to closely related surface species. The same is true for satellite DNA.

These analyses will also speak to the authors' interest in gene family evolution bc extreme population bottlenecks at the dawn of cave colonization may be an explanation for the exceptional gene family size changes they detected in the cave lineage MRCAs.

> Authors: We agree with the reviewer 2 that non-coding elements are of great importance to understand adaptive evolution as seen in cavefish (Krishnan et al. 2022). Nevertheless, as we are using transcriptomes instead of genomes, we do not have information about regulatory, mobile elements, inversions, or other genomic changes that could be involved in subterranean adaptation. However, we do know that gene gain, duplication and loss are key drivers of adaptation and since we can address these events with the use of highly complete transcriptomes, we focused on the evolutionary dynamics of the coding-gene repertoire. We have changed our phrasing to be more explicit about the type of data that we analyze: "Here, we explore the genomic underpinnings of adaptations to life in caves in Coleoptera using a genome-wide phylogenomic approach to investigate and characterize gene repertoire evolutionary dynamics across surface-dwelling and subterranean species that represent multiple independent underground colonizations both in terrestrial (Leptodirini) and aquatic beetle lineages (Hydroporini and Bidessini)". Also we further acknowledge the limitations of our study in this regard in the discussion section.

Another crucial genome partition to look at are pseudogenes as the most recent direct evidence of processes in the process of regressive evolution. This data source is one of the most direct ones regarding the relative contributions of regressive vs constructive trait change in cave adaptation, which is a major question in the field at this point.

Recent work by the Cooper lab nicely pinpointed vision-related pseudogenes in the subterranean diving beetle genera included in this submission (Langille et al., 2022). From the perspective of sequence quality assessment as well as evolutionary informativeness, it seems an obvious question whether these were captured in the new genome sequence drafts.

> Authors: We agree with the reviewer 2, pseudogenization has been described as a remarkable evolutionary process in cave-dwelling fauna in particular gene families, such as phototransduction genes in Dytiscidae (Langille et al., 2022). Our research is designed to detect global patterns of gene repertoire changes, a major evolutionary force, related to independent underground transitions. In this context, pseudogenization is only one of the several mechanisms that can lead to gene loss (Albalat & Cañestro 2016) and have thus been indirectly taken into account in our study when exploring gene family contractions. Also note that at this stage most of our data comes from transcriptomes, thus we do not have the whole genome sequences to investigate the pseudogenization process directly. In our opinion, future research on particular gene families at a shallower evolutionary scale should also include an estimation of pseudogenization processes to understand the evolutionary mechanisms underlying adaptation through gene loss and their impact on

subterranean-related phenotypes. Nonetheless, we acknowledge the reviewer's point and mention this while discussing the limitations and future prospects of our study.

In extension, it should be noted that even actual gene loss events could be pinpointed and quantified with high confidence through the comparative analysis of synteny traces.

> Authors: We agree with the reviewer, but as explained above, please note that we have no chromosome-level genomic information of these species, and thus we are still not at the point of exploring synteny traces. In the new version of the manuscript, we further clarify the type of data that we based our analysis on: "These datasets contain 22 species from the tribes Leptodirini, Hydroporini, and Bidessini, including 21 newly sequenced transcriptomes for this study...".

In light of these gaps, I think the "mainly" in the statement "Our results indicate that remarkable genomic changes mainly driven by gene family expansions ..." is not sufficiently supported at this point. Phenotypic evolution results from both cis-regulatory changes and protein sequence changes besides gene family size changes (Stern, 2011). At this point, we study does not provide insights into the relative contribution of cis-regulatory DNA change to cave adaptation in their lineages.

> Authors: We agree with the reviewer. We have now substituted "genomic changes" by "gene repertoire changes" and acknowledge that while we are only focusing in the evolutionary dynamics of the coding-gene repertoire, other sources of genomic variation could be indeed involved as well in adaptation to life in caves.

2.

"Our results thus indicate that the MRCAs of the three tribes experienced a significant level of gene gain and duplication, and suggest that genomic exaptation fueled by an expansion of the gene repertoire may have paved the way for a later colonization of the underground in these three tribes."

The authors adequately define their gene exaptation model as a possibility at this point. I would agree that it is not yet possible to conclude with high confidence that "exaptation" of duplicated genes for cave traits has taken place. For this, we would need to know about their functionality as nascent gene duplicates and how this ancestral function compares to that in cave-adapted descendants. The authors should consider making attempts in this direction by looking at some well-understood genes in their data set.

While I consider the analysis of gene family size change histories as thorough and highly reliable, it is important to remember that gene duplication outcomes can be phenotypically neutral (Lynch & Conery, 2000). In extension, GO-term enrichment in expanded gene family populations can result from both adaptive effects or tolerance of gene dosage increase. Selection tests are required to distinguish these very different scenarios. This can be as straightforward as relative rate tests or dN/dS. I think this is a compulsory additional analysis the authors need to carry out bc it has a high potential to clarify the adaptive significance of the duplicated gene partition. The opportunity to explore paralog diversification is a present that comes along with gene duplications.

> Authors: We agree with the reviewer in that testing selection in expanded gene families would provide an alternative perspective to our research. However, we do think that both copy-number variation and point mutations form the basis for most evolutionary innovation and are indeed different processes, as recently discussed in Tomanek and Guet (2022). In order to follow the reviewer's recommendation, we have now tested positive selection in the expanded OGs in parallel in the highly modified lineages of the tribe Leptodirini (Fig. 5). We have done this to provide an example of to what extent these parallel expansions are further impacted with positive selection and are potentially related to the extreme subterranean adaptations that these independent lineages developed. We found a remarkable proportion of them with signatures of positive selection, suggesting that these expansions may have both effects in gene dosage increase and, in some cases, potentially diverged to alternative functions (i.e., neofunctionalization, subfunctionalization or multifunctionalization). This alternative test is now included in the new version of the manuscript.

Also note that while there is rationale to consider exaptation and cooption equivalent terms, exaptation in a strict sense does not speak to whether a new gene function completely replaced an ancestral one, while cooption unambiguously implies the conservation of ancestral functionality during the acquisition of new additional functions. Both scenarios are conceivable. My understanding of the literature is that cooption is the far more frequently documented trajectory. So the authors should include this distinction and ideally test for them for a few genes with well-documented functions.

> Authors: The reviewer has brought up an important point to discuss. Indeed, exaptation refers to a trait or feature that evolved for one purpose but has later become useful for a different one. In other words, it is the process by which a structure or trait evolves for one function and is later used for a completely different one. On the other hand, co-option refers to the process by which a trait or a feature that evolved for one purpose is then used for a different one without necessarily evolving any further. In other words, it is the process of taking an existing trait or feature and putting it to use for a new function. In essence, exaptation is the process of evolutionary change, whereas co-option is a process of using existing traits for new purposes (Gould and Vrba 1982; Stern 2013). Comparing the same trait (in the case of this study, gene family evolution) across different lineages provides insights for whether the trait evolved independently in different lineages for the same function (exaptation) or if it evolved once and then was co-opted in different lineages (co-option). In our case, since the trait (e.g., gene family expansion) evolved differently in three different beetle lineages (e.g., gene gain and duplication in the MRCA of Leptodirini, Hydroporini and Bidessini), we consider that the most correct term to use would be exaptation, but agree with the reviewer in that using either term with robustness is complicated.

3.

“Adaptation to life in caves is often accompanied by dramatically convergent changes across distantly related taxa, epitomized by the loss or reduction of eyes and pigmentation. Nevertheless, the genomic underpinnings underlying the evolution of cave-related phenotypes are largely unexplored.”

Following up on this statement, the authors do cite previous work that reported pivotal genetic changes underlying cave adaptive trait changes. However, I do feel this presentation falls short of giving adequate oversight on the essential insights gained through the genetic and genome-wide approaches in cavefish. It will be necessary to fill this gap to be able to illustrate the general significance of their new work in the discussion section.

> Authors: We have now modified this statement in the abstract to highlight the scarcity of genome-wide macroevolutionary studies on cave fauna. To our knowledge, there are very few published studies and all of them have focused on vertebrates (e.g., Policarpo et al. 2021, Li et al. 2021, Bondareva et al. 2023). We initially cited several genomic studies in particular cave-dwelling species in the introduction and included some examples of cave-fish. We have now substituted some of the previously cited studies with more recent ones. The list of examples could be extended, but we have opted to only cite a selection of recent studies as to not overpass the maximum number of citations allowed in this journal, especially given the additional citations we have been asked to include by the reviewers.

4.

“This parallel pulse of gene gain and loss in independent highly specialized subterranean lineages supports the idea of life in caves not as a ‘wreck of ancient life’, as defined by Darwin, but actually as an opportunity for genomic innovation through gene family expansion.”

+

“we show that this interpretation of the phenotype should be better defined as a reshaping of the gene repertoire (and eventually of the phenotype) mainly driven by gene gain. Darwin’s ‘wrecks of ancient life’ also rise, after all.”

Given the lack of functional insights at this point, I don’t think it is clear how much the investigated cave beetle lineages are truly genetically more complex compared to the MRCAs with their surface relatives. Independent of the answer, it would not change the previously characterized ratio of regressive vs constructive trait changes at the level of the phenotype.

> Authors: We do not describe an increase of complexity of the gene repertoire associated with subterranean evolution, instead we found that both subterranean and surface-dwelling lineages have experienced substantial changes in their gene repertoires compared to their MRCAs. Through these points, the idea we want to highlight is precisely that cave adaptation is not only epitomized by a “gene repertoire simplification” driven by gene loss (and pseudogenization), but is encompassed by reshaping of the gene repertoire fueled by both gene loss and gene gain, with this last event being surprisingly prevalent compared to our previous knowledge about subterranean evolution from the molecular perspective.

Finally, as mentioned above, the question of how much adaptive vs non-adaptive processes are part of cave-adaptive transitions remains one of the core questions in the field. The authors understandably and convincingly pinpoint the parallel expansions of gene repertoires in their study. They also emphasize the important finding that these expansions

preceded the actual process of cave colonization. In my mind, an intriguing non-adaptive explanation for these findings is the occurrence of extreme population size shrinkages of the source lineages that managed to persist before or in the process of retreating into cave ecologies. However, other scenarios are conceivable. Only further insights into the adaptive vs non-adaptive outcomes of the detected gene duplications have the high potential to provide functional as opposed to corollary insights.

> Authors: We completely agree with the reviewer. Further studies on this interesting topic are definitely most needed.

Minor issues:

1.

“Furthermore, gene gains and duplications were more abundant...”

Does the listing of “gene gain” next to “duplication” refer to gene duplication-independent processes?

> Authors: Gene duplication is a particular type of gene gain, but gene gain can also occur through alternative evolutionary events. In order to avoid confusion, we have modified this sentence by removing the word “duplication”.

2.

“genome-wide phylogenomic approach”

This seems redundant to me. Phylogenomic implies genomic

> Authors: We respectfully disagree with the reviewer. Phylogenomic implies genomic, but it does not imply analyzing genome-wide information (e.g. one may analyze 1,000 gene families and will be still using phylogenomics). Therefore, we prefer to keep both terms.

3.

The category “compound eye development” is captured in GO terms that both contracted and expanded in parallel in Fig. 3c. It would be helpful if these genes are explored for an explanation of these opposite trends.

> Authors: We agree that this is an interesting point that could be further explored. We used Gene Ontology (GO) annotations to simplify and provide a general perspective of the putative functions of the expanded and contracted gene families in our study. In this case, we collected the deepest GO term for each OG to represent the most precise functions of the GO hierarchy (“reduced GO term”, explained in the methods). However, the candidate functions of these gene families should be further investigated, especially when we observe these contrasting patterns in processes such as “eye development”, the most studied process in cave fauna from the molecular perspective. We have now explored more the putative functions of the gene families that have expanded and contracted in more depth

based on all the annotations we obtained through EggNOG-mapper, and proposed a few hypotheses about the biological meaning of our findings.

The gene family annotations of the parallelly contracted OGs correspond to essential components for eye development in *Drosophila*, such as for MYO5A, RBBP6, OPA1, and FZR1 (Sato et al. 2008; Hull et al. 2015; Yarosh et al. 2008; Martins et al. 2017). In contrast, the parallelly expanded OGs correspond to other gene families related to eye development as well as other functions in the nervous system such as SDK2, ATXN2, boss, EYS among others. For instance, EYS (eyes shut / spam) plays a key role in mechanosensory/chemosensory neurons by preserving cell shape under environmental stress (Cook et al. 2008) and its expression pattern has been found to alternatively be co-opted and expanded from mechanosensory/chemosensory neurons to photoreceptor cells in insects with open rhabdoms such as *Drosophila* (Mahato et al. 2018). The parallel expansions observed here could be related to modifications of the extra-optic sensory systems in a long-term subterranean evolution scenario. Nevertheless, we cannot address a more exhaustive interpretation of these results unless we perform functional experiments on our study system, which are definitely necessary to understand the eye loss mechanisms in cave beetles.

Interestingly, these events of parallel expansion and contraction of gene families involved in compound eye development were mainly detected between the subterranean lineages of the aquatic clade, whose surface-dwelling relatives have fully functional eyes. Only one lineage of the terrestrial tribe (L3) shows a parallel expansion with B1 (OG0001321), and its annotation corresponds to ATXN2. An overexpression of ATXN2 (Ataxin-2) has been shown to cause retinal degeneration (Elden et al. 2010), indicating that possibly an expansion of this gene family is related to negative regulation of eye development in cave beetles. In summary, loss of eyes and all the neural circuits associated with this complex structure, as observed in some of these cave-beetle species (Luo et al. 2019), might therefore be caused by pseudogenization and ultimately loss of genes involved in eye development but also by expansions of gene families putatively involved in their regulation and alternative functions in the nervous system development. We have added these results to the Supplementary Data 4 and briefly discussed them in the new version of the manuscript.

4.

“In the context of this study, we refer to parallel evolution as significant shifts in net rates occurring in gene families shared by all or several lineages. In contrast, we define convergent evolution as lineage-specific significant changes in net rates in gene families that have similar putative functions when compared across lineages.”

(Scotland, 2010) should be cited for a clear definition of parallel vs convergent which matches that of the authors.

> Authors: We have now added this citation.

Langille, B. L., Tierney, S. M., Bertozzi, T., Beasley-Hall, P. G., Bradford, T. M., Fagan-Jeffries, E. P., Hyde, J., Leijs, R., Richardson, M., Saint, K. M., Stringer, D. N., Villastrigo, A., Humphreys, W. F., Austin, A. D., & Cooper, S. J. B. (2022). Parallel decay of

vision genes in subterranean water beetles. *Molecular Phylogenetics and Evolution*, 173, 107522.

Lynch, M., Bobay, L.-M., Catania, F., Gout, J.-F., & Rho, M. (2011). The repatterning of eukaryotic genomes by random genetic drift. *Annual Review of Genomics and Human Genetics*, 12, 347–366.

Lynch, M., & Conery, J. S. (2000). The evolutionary fate and consequences of duplicate genes [Review of The evolutionary fate and consequences of duplicate genes]. *Science*, 290(5494), 1151–1155. [sciencemag.org](https://www.sciencemag.org).

Lynch, M., & Conery, J. S. (2003). The origins of genome complexity. *Science*, 302(5649), 1401–1404.

Scotland, R. W. (2010). Deep homology: a view from systematics. *BioEssays: News and Reviews in Molecular, Cellular and Developmental Biology*, 32(5), 438–449.

Stern, D. L. (2011). *Evolution, Development, and the Predictable Genome*. W. H. Freeman.

References:

Krishnan, J. *et al.* Genome-wide analysis of cis-regulatory changes underlying metabolic adaptation of cavefish. *Nat. Genet.* **54**, 684–693 (2022).

Albalat, R. & Cañestro, C. Evolution by gene loss. *Nat. Rev. Genet.* **17**, 379–391 (2016).

Tomanek, I. & Guet, C. C. Adaptation dynamics between copy-number and point mutations. *Elife* **11**, (2022).

Gould, S. J. & Vrba, E. S. Exaptation—a Missing Term in the Science of Form. *Paleobiology* vol. 8 4–15 Preprint at <https://doi.org/10.1017/s0094837300004310> (1982).

Stern, D. L. The genetic causes of convergent evolution. *Nat. Rev. Genet.* **14**, 751–764 (2013).

Policarpo, M. *et al.* Contrasting Gene Decay in Subterranean Vertebrates: Insights from Cavefishes and Fossorial Mammals. *Mol. Biol. Evol.* **38**, 589–605 (2021).

Li, R. *et al.* Whole-Genome Sequencing of *Sinocyclocheilus maitianheensis* Reveals Phylogenetic Evolution and Immunological Variances in Various *Sinocyclocheilus* Fishes. *Front. Genet.* **12**, 736500 (2021).

Bondareva, O. *et al.* How Voles Adapt to Subterranean Lifestyle: insights from RNA-seq. *Frontiers in Ecology and Evolution* **11**, 105 (2023).

Satoh, A. K., Li, B. X., Xia, H. & Ready, D. F. Calcium-activated Myosin V closes the *Drosophila* pupil. *Curr. Biol.* **18**, 951–955 (2008).

Hull, R. *et al.* The *Drosophila* retinoblastoma binding protein 6 family member has two isoforms and is potentially involved in embryonic patterning. *Int. J. Mol. Sci.* **16**, 10242–10266 (2015).

Yarosh, W. *et al.* The molecular mechanisms of OPA1-mediated optic atrophy in *Drosophila* model and prospects for antioxidant treatment. *PLoS Genet.* **4**, e6 (2008).

Martins, T., Meghini, F., Florio, F. & Kimata, Y. The APC/C Coordinates Retinal Differentiation with G1 Arrest through the Nek2-Dependent Modulation of Wingless Signaling. *Dev. Cell* **40**, 67–80 (2017).

Cook, B., Hardy, R. W., McConnaughey, W. B. & Zuker, C. S. Preserving cell shape under environmental stress. *Nature* **452**, 361–364 (2008).

Mahato, S., Nie, J., Plachetzki, D. C. & Zelhof, A. C. A mosaic of independent innovations involving eyes shut are critical for the evolutionary transition from fused to open rhabdoms. *Dev. Biol.* **443**, 188–202 (2018).

Elden, A. C. *et al.* Ataxin-2 intermediate-length polyglutamine expansions are associated with increased risk for ALS. *Nature* **466**, 1069–1075 (2010).

Luo, X.-Z., Antunes-Carvalho, C., Wipfler, B., Ribera, I. & Beutel, R. G. The cephalic morphology of the troglobiontic cholevine species *Troglocharinus ferreri* (Coleoptera, Leiodidae). *J. Morphol.* **280**, 1207–1221 (2019).

Reviewer #3 (Remarks to the Author):

Overall, this paper is well-written and interesting. It is likely to have a notable and positive impact on insect/beetle comparative genomics, including our understanding of the evolution of cave-dwelling habits. While there have been many studies of cave-dwelling insects, including beetles, none to my knowledge have undertaken a comparative genomic study involving multiple distantly-related species with convergent habits. The choice of taxa is interesting because it spans the basal split in Coleoptera and the reconstructed phylogeny for Coleoptera is consistent with the best available estimates from other studies. Most of the authors conclusions are based on homology to genes of known function in distantly-related species and many conclusions are generalizations, leaving much uncertainty. However, I cannot conceive of a suitable remedy for this at the taxonomic scale at which they are working and I consider the results both interesting and important. Their results suggest many intriguing paths for future study, including functional validation/characterization of genes proposed to be involved in specializations for cave life, notably including metabolism, vision, and development. The methodology is sound and meets the expected standards in phylogenetic systematics and comparative genomic analyses. I have only a few suggestions (see below) for improvement of the text, which is generally very well-written. Note, line numbering would have been helpful.

> Authors: We thank the reviewer for their comments and for their interest in our findings.

Introduction:

The first sentence appears to be an incomplete thought. Rewrite for clarity.

> Authors: We thank the reviewer for noticing. We have corrected this and included the full sentence.

Page 2: The following sentence is unclear/incomplete: Diving beetles (Dytiscidae) also represent a large radiation in subterranean aquatic ecosystems; they colonized independently multiple underground desert aquifers in Western Australia being this colonization 'recent' (ca. 12 - 9 Mya respectively).

> Authors: We have corrected the sentence with the estimated subterranean transitions in the diving beetle tribes: 'Diving beetles of the tribes Bidessini and Hydroporini (Dytiscidae, Hydroporinae) also represent large radiations in subterranean aquatic ecosystems; they colonized independently multiple underground desert aquifers in Western Australia ca. 7–3 Mya^{24,25}.'

Page 2:

It would be helpful if you indicated here the beetle families to which these tribes belong, as well as their estimated divergence times from other taxa studied.

> Authors: We have now added this information.

Results & Discussion:

Line 1: The statement “the main beetle lineages plus two outgroups” is too vague. What lineages were sampled (suborders, series or superfamilies; were any series or superfamilies missing?) and what outgroup taxa were sampled? This is elsewhere in your paper/supplement, but is not easy/quick to find nor apparent in the labeling of your phylogeny.

> Authors: We have now clarified the beetle lineages and outgroups included in our study. It now reads ‘the main beetle lineages (Myxophaga, Archostemata, Adephaga and Polyphaga) plus two outgroups (Neuroptera and Strepsiptera), in order to explore gene repertoire evolution in a broad phylogenetic framework that encompass the lineages of interest”.

The phylogeny shown in Fig 1a includes all the Coleoptera, Strepsiptera, and Neuroptera species included in our study. We only labeled the superfamily, subfamily and tribe names of the lineages of interest to highlight them because we used these names throughout the text. Yes, there are superfamilies not represented in this phylogeny. For some of the initially selected superfamilies we did not find public data, or when we did their quality of the transcriptomes/genomes was too low. We successfully collected data for some of the flanking lineages to our groups of interest, allowing us to explore gene repertoire evolution in these particular clades and sister groups. As you can observe, we included several staphylinoids and ditiscoids species based on (i) their availability in public repositories and (ii) the quality of their transcriptomes/genomes based on the BUSCO analysis, therefore providing more information about the ancestral states of tribes of interest.

Need commas after “i.e.” and “e.g.” throughout.

> Authors: Corrected.

Figs 4a 4b too cluttered and complicated. Informative, but overwhelming to look at. I cannot immediately recommend a good solution to this, but I hope you will consider rethinking how these data are presented to both make it easier for the reader to interpret and to avoid overwhelming the reader with too much information. At a minimum, I recommend directing the reader (in the legend) to your most notable results.

> Authors: We agree with the reviewer, but we consider both heatmaps and constellation plots indispensable elements to understand the degree of functional convergence at a global and lineage-specific levels respectively. We have now modified the color of the functional clusters with the same annotation in both Fig. 4a and 4b, thus simplifying its interpretation. We have also rewrote the legend, highlighting the most notable findings reflected in the figure.

Methods:

I would rethink citing Cai et al. (2022) for its topology. There are problems with the analyses in that paper and the subordinal relationships it recovered differ from those in your paper as well as other recent phylogenomic studies. See Boudinot et al. (in press Syst. Ent.; <https://doi.org/10.1111/syen.12570>) for more information. Nonetheless, I think it is reasonable to cite Cai et al. 2022 for fossils.

> Authors: We only cited Cai et al. (2022) for fossil calibration points. Our topology overlaps with the currently accepted phylogenetic relationships of the Coleoptera superfamilies and we cited McKenna et al. (2019) for its topology, which was used to revise the non-supported node: “All nodes had more than 90% of bootstrap support except the split between *Onthophagus taurus* and *Hydrochus megaphallus* (i.e., 55%) (Supplementary Fig. 6a). We followed the topology of McKenna et al. (2019)⁴¹ (i.e., using a phylogenomic approach with a broader taxa representation) for the placement of these taxa.”

Reviewers' Comments:

Reviewer #1:

Remarks to the Author:

The authors have carefully revised their manuscript, and I do not have further comments to make. Nice work!

Reviewer #2:

Remarks to the Author:

Major issues:

1.

Authors: "Here, we explore the genomic underpinnings of adaptations to life in caves in Coleoptera using a genome-wide phylogenomic approach."

Reviewer: I sincerely apologize for missing the fact that the study is based on 21 newly generated transcriptomes, 13 previously published transcriptomes, and 10 previously published genome sequence drafts. A significant reason causing this unfortunate oversight, however, is the fact that these basic numbers are not provided in the main text. Moreover, the term "genomic" is used 45 times, while "transcriptomic" and "exonic" are used 4 and 0 times, respectively. Equally important, while transcriptome-based phylogenies approximate truly phylogenomic analyses sufficiently enough to justify equivalent use of the terms, the original meaning of "phylogenomic" referred to inferences from genomic datasets (<https://en.wikipedia.org/wiki/Phylogenomics>). I therefore strongly recommend the authors replace "genomic" with "transcriptomic" or "transcriptome-based" wherever conducive in the manuscript. An equally important and effective improvement would be to revise the abstract by adding that the study is critically based on the 21 new transcriptomes. This is an important contribution to the field all by itself.

Note that the comment by reviewer 1 on the completeness of the primary data measured by BUSCO scores is affected by the nature of the primary data. While transcriptome data from one life history state are an effective way to approach genomic coverage of gene content, they are, by nature only an approximation. The authors are to be credited for their scrutiny to subject their primary data to BUSCO analysis. However, my advice is to revise the manuscript to the effect that some of the inherent limitations of the transcriptome resources are pointed out earlier in the manuscript.

Ideally, the authors should begin the results section with a paragraph that describes the acquisition and evaluation of the newly generated transcriptome data. This continues to be the standard narrative for genome papers. It does set the stage for evaluating all the analyses downstream of the raw data in an adequate manner. In this new section, the authors should also state that they were consistent in generating adult whole-body transcriptomes. They could also point out that embryonic transcriptomes are virtually impossible to generate in this case bc of the extremely slow reproductive rate of the reclusive organisms they work with.

Finally, please note that the quantification of gene losses is more sensitive to the limitations of the transcriptomic approach than the quantification of relative gene family expansions. This is bc of the basic difference between positive and negative evidence. Gene loss inferences are based on negative evidence, which can only be preliminary if genome content is sampled in an approximative manner through transcriptome data. It would be useful if the authors could address this in the discussion.

Authors: " In essence, exaptation is the process of evolutionary change, whereas co-option is a

process of using existing traits for new purposes (Gould and Vrba 1982; Stern 2013). Comparing the same trait (in the case of this study, gene family evolution) across different lineages provides insights for whether the trait evolved independently in different lineages for the same function (exaptation) or if it evolved once and then was co-opted in different lineages (co-option). In our case, since the trait (e.g., gene family expansion) evolved differently in three different beetle lineages (e.g., gene gain and duplication in the MRCA of Leptodirini, Hydroporini and Bidessini), we consider that the most correct term to use would be exaptation, but agree with the reviewer in that using either term with robustness is complicated.”

Reviewer: I agree with this discussion of the terms exaptation and co-option. I don't think, however, that "gene family expansion" is a functional trait. It is a quantitative trait of the coding genome. As such, it does only allow minimal functional inferences. I, therefore, continue to think that co-option is the safer term at this point. Alternatively, in the discussion, "co-option and possibly exaptation" could be used.

Minor issues:

2.

"genome-wide phylogenomic approach"

This seems redundant to me. Phylogenomic implies genomic

> Authors: We respectfully disagree with the reviewer. Phylogenomic implies genomic, but it does not imply analyzing genome-wide information (e.g. one may analyze 1,000 gene families and will be still using phylogenomics). Therefore, we prefer to keep both terms.

Reviewer: I fully respect the authors' authority to make a final decision on this, but I feel obliged to state that it continues to be difficult for me to understand their reasoning. Moreover, given the potentially misleading preponderance of the "genomic" vs "transcriptomic" in the current manuscript version pointed out above, I encourage the authors to take one more look at what might be better language here.

3.

"The gene family annotations of the parallelly contracted OGs correspond to essential components for eye development in *Drosophila*, such as for MYO5A, RBBP6, OPA1, and FZR1 (Satoh et al. 2008; Hull et al. 2015; Yarosh et al. 2008; Martins et al. 2017)."

Reviewer: I appreciate that the authors added more specific information on the eye development-related genes. This is effective in helping to understand the nature of the primary evidence that leads to the GO term trend findings. The authors should note, however, that all four genes code for highly pleiotropic products. While visual system-related lack of function phenotypes have been described, these are the results of broader cellular functions instead of uniquely visual system-related functions.

At the same time, I continue to think that a section describing the recovery of opsin transcripts in relation to eye preservation in the different species represents an essential addition to the manuscript.

Reviewer #3:

Remarks to the Author:

My concerns have been addressed in the revisions.

REVIEWER COMMENTS

Reviewer #1 (Remarks to the Author):

The authors have carefully revised their manuscript, and I do not have further comments to make. Nice work!

> Authors: We thank the reviewer for their constructive comments on the initial version of the manuscript, which have helped to substantially improve it.

Reviewer #2 (Remarks to the Author):

Major issues:

1.

Authors: “Here, we explore the genomic underpinnings of adaptations to life in caves in Coleoptera using a genome-wide phylogenomic approach.”

Reviewer: I sincerely apologize for missing the fact that the study is based on 21 newly generated transcriptomes, 13 previously published transcriptomes, and 10 previously published genome sequence drafts. A significant reason causing this unfortunate oversight, however, is the fact that these basic numbers are not provided in the main text. Moreover, the term “genomic” is used 45 times, while “transcriptomic” and “exonic” are used 4 and 0 times, respectively. Equally important, while transcriptome-based phylogenies approximate truly phylogenomic analyses sufficiently enough to justify equivalent use of the terms, the original meaning of “phylogenomic” referred to inferences from genomic datasets (<https://en.wikipedia.org/wiki/Phylogenomics>). I therefore strongly recommend the authors replace “genomic” with “transcriptomic” or “transcriptome-based” wherever conducive in the manuscript. An equally important and effective improvement would be to revise the abstract by adding that the study is critically based on the 21 new transcriptomes. This is an important contribution to the field all by itself.

>Authors: We apologize for the lack of clarity, and have now stated several times throughout the manuscript that the source of the newly generated material is transcriptomes and that we analyzed a set of proteomes generated through a combination of genomes and transcriptomes. Regarding the term phylotranscriptomics, we are not completely sure it is the best term to refer to our analyses, since as mentioned above our analyses contain both genomic and transcriptomic data. In addition, we believe that the original conception of the word phylogenomics as coined by Jonathan Eisen in his paper 'Phylogenomics: Improving Functional Predictions for Uncharacterized Genes by Evolutionary Analysis' (1998) was related to phylogenetic analyses of multiple genes, regardless of them being annotated from a genome or a transcriptome ('I present an outline of one such *phylogenomic* method (see Fig. 1), and I compare this method to nonevolutionary functional prediction methods. This method is based on

a relatively simple assumption—because gene functions change as a result of evolution, reconstructing the evolutionary history of genes should help predict the functions of uncharacterized genes. The first step is the generation of a phylogenetic tree representing the evolutionary history of the gene of interest and its homologs (...). Therefore, we believe that based on the original meaning of the word, it is correct to use it in the context of our study. In any case, we highlight that we have worked throughout the manuscript to add clarity about our proteomes being inferred from transcriptomes and not from genomes, which we understand is the key aspect of the discussion around this point raised by the reviewer.

Note that the comment by reviewer 1 on the completeness of the primary data measured by BUSCO scores is affected by the nature of the primary data. While transcriptome data from one life history state are an effective way to approach genomic coverage of gene content, they are, by nature only an approximation. The authors are to be credited for their scrutiny to subject their primary data to BUSCO analysis. However, my advice is to revise the manuscript to the effect that some of the inherent limitations of the transcriptome resources are pointed out earlier in the manuscript.

Ideally, the authors should begin the results section with a paragraph that describes the acquisition and evaluation of the newly generated transcriptome data. This continues to be the standard narrative for genome papers. It does set the stage for evaluating all the analyses downstream of the raw data in an adequate manner. In this new section, the authors should also state that they were consistent in generating adult whole-body transcriptomes. They could also point out that embryonic transcriptomes are virtually impossible to generate in this case bc of the extremely slow reproductive rate of the reclusive organisms they work with.

> Authors: All publicly available genomic and transcriptomic data together with the newly generated ones were subjected to a strict quality control to ensure their high quality, not only in terms of BUSCO completeness, but also contamination filtering. The datasets are included in our MATEdb database, where the readers can find the full details of how the curation of the datasets was done. In order to avoid providing extensive information that can be found somewhere else, we have opted for citing the MATEdb manuscript while providing a summary of the steps followed. Please note that the debate about quality check applies to both transcriptomic and genomic dataset (as discussed more in depth below), and since this manuscript investigates gene repertoire evolution, we carefully checked it in both types of datasets. As suggested, we have added information about the source for the transcriptomes and the unavailability of embryonic transcriptomes due to the difficulty in finding them.

Finally, please note that the quantification of gene losses is more sensitive to the limitations of the transcriptomic approach than the quantification of relative gene family expansions. This is bc of the basic difference between positive and negative evidence. Gene loss inferences are based on negative evidence, which can only be preliminary if genome content is sampled in an approximative manner through transcriptome data. It would be useful if the authors could address this in the discussion.

> Authors: We believe there is a misconception in the scientific community about a genome being 'better' than a transcriptome for gene repertoire evolution inferences. This is far from true. We would like to respectfully argue, if we may, that both transcriptomic and genomic approaches are subject to methodological and analytical bias, that the quality of the genomes (and hence, the robustness of the inferences that we can make with them) vary enormously depending on the type of data and software used for assembly and annotation, and that most genomes are in fact at the stage of draft genomes (eg, it is very common that the total number of genes is underestimated as tandem duplicated genes may be 'collapsed' under the same sequence instead of recovered as such, or that draft assemblies result in an overestimation of the number of genes as they may recover the same gene splitted in several scaffolds). Thus, we think that the comparison between transcriptomes and genomes is not so straightforward, and we believe that the inclusion of genomes instead of transcriptomes in gene repertoire evolution studies could potentially add more noise than signal if the genome is not of high quality, and hence it is not necessarily better than including only transcriptomes. We therefore argue that it is not the source of the inferred proteome what matters (ie, annotated genes coming from a genome or a transcriptome), but their *quality*. For instance, the proteome annotated in a chromosome-level genome will be far from complete or correct if low quality transcriptomes or transcriptomes of different species are used for its annotation. It is even worse in the case of draft genomes, where the fragmentation can strongly affect the proper annotation of genes that may be present in the transcriptome of the same species. It is for this reason that our emphasis is on checking that we only have high quality genomes and transcriptomes, because we believe that it is the best way to minimize biases associated with downstream analysis of copy number variation. Regarding the latter, as wisely pointed out by the reviewer, low quality datasets (either transcriptomic or genomic) can result in an inflated inference of gene losses, hence again our emphasis of only keeping high quality datasets. In this context, it has been extensively proven that phylotranscriptomics is as accurate as phylogenomics for gene repertoire evolution studies as far as their quality is good (eg, Cheon et al. 2020. Is phylotranscriptomics as reliable as phylogenomics? Mol. Biol. Evol.; Fernández & Gabaldón 2020. Gene gain and loss across the metazoa tree of life. Nat. Ecol. Evol.), and therefore we find that it might not be necessary to add this information in this piece of work.

Authors: " In essence, exaptation is the process of evolutionary change, whereas co-option is a process of using existing traits for new purposes (Gould and Vrba 1982; Stern 2013). Comparing the same trait (in the case of this study, gene family evolution) across different lineages provides insights for whether the trait evolved independently in different lineages for the same function (exaptation) or if it evolved once and then was co-opted in different lineages (co-option). In our case, since the trait (e.g., gene family expansion) evolved differently in three different beetle lineages (e.g., gene gain and duplication in the MRCA of Leptodirini, Hydroporini and Bidessini), we consider that the most correct term to use would be exaptation, but agree with the reviewer in that using either term with robustness is complicated."

Reviewer: I agree with this discussion of the terms exaptation and co-option. I don't think, however, that "gene family expansion" is a functional trait. It is a quantitative trait of the coding

genome. As such, it does only allow minimal functional inferences. I, therefore, continue to think that co-option is the safer term at this point. Alternatively, in the discussion, “co-option and possibly exaptation” could be used.

> Authors: We agree with the reviewer in that including both terms may be the wisest option, given that proving with robustness either one of the other is very challenging in this case. We have changed ‘exaptation’ by ‘either exaptation or co-option’ throughout the main section and conclusions.

Minor issues:

2.

“genome-wide phylogenomic approach”

This seems redundant to me. Phylogenomic implies genomic

> *Authors (response to first submission)*: We respectfully disagree with the reviewer. Phylogenomic implies genomic, but it does not imply analyzing genome-wide information (e.g. one may analyze 1,000 gene families and will be still using phylogenomics). Therefore, we prefer to keep both terms.

>Authors: We have now deleted ‘genome-wide’.

Reviewer: I fully respect the authors’ authority to make a final decision on this, but I feel obliged to state that it continues to be difficult for me to understand their reasoning. Moreover, given the potentially misleading preponderance of the “genomic” vs “transcriptomic” in the current manuscript version pointed out above, I encourage the authors to take one more look at what might be better language here.

> Authors: Since our analysis is based on a combination of genomic and transcriptomic datasets, we believe the term phylogenomics is still more accurate than phylotranscriptomics. Nonetheless, we agree with the reviewer in that we should add clarity about the source of the datasets analyses, and have taken action along these lines on the revised version of the manuscript. We have now clarified throughout the manuscript that we are using a combination of genomes and transcriptomes for the phylogenomic analyses, and emphasize that in this manuscript we investigate gene repertoire evolution.

3.

“The gene family annotations of the parallelly contracted OGs correspond to essential components for eye development in *Drosophila*, such as for MYO5A, RBBP6, OPA1, and FZR1 (Sato et al. 2008; Hull et al. 2015; Yarosh et al. 2008; Martins et al. 2017).”

Reviewer: I appreciate that the authors added more specific information on the eye development-related genes. This is effective in helping to understand the nature of the primary evidence that leads to the GO term trend findings. The authors should note, however, that all four genes code for highly pleiotropic products. While visual system-related lack of function phenotypes have been described, these are the results of broader cellular functions instead of uniquely visual system-related functions.

At the same time, I continue to think that a section describing the recovery of opsin transcripts in relation to eye preservation in the different species represents an essential addition to the manuscript.

> Authors: We thank the reviewer for appreciating the new information added to the manuscript. We agree with the reviewer in that adding more information on the topic would strengthen the paper. However, we believe that it is challenging to have a deeper understanding on the matter without the generation of genomic data (eg, chromosome-level genomes) and organ-specific transcriptomic data, which is out of scope for the current piece of work. We have, however, extended the discussion about opsin genes based on previous studies of some of the coauthors of this piece of work (Langille et al. 2022. Parallel decay of vision genes in subterranean water beetles, Mol Phyl Evol), where it was reported that opsin and other phototransduction genes in 32 subterranean species show loss of function mutations and there is a trend across all the subterranean dytiscid species for parallel loss of these genes related to vision. In addition, we have added a note of caution specifying that some of the genes with parallel changes code for pleiotropic proteins and that therefore further functional studies would be necessary to validate our findings and further understand the genomic underpinnings of eye regression in these cave beetles. Accordingly, the following sentence has been added: *'These results are in line with the findings of previous works¹⁹ that reported evidence of parallel coding sequence decay in eight phototransduction genes in 32 subterranean species in contrast to surface species as a consequence of the accumulation of frameshift mutations and premature stop codons. However, given the pleiotropic nature of some of these genes (eg, MYO5A, RBBP6, OPA1, FRZ1) and the mechanistic complexity involved in eye reduction in other cave organisms involving non-coding genomic elements and dynamics⁷⁵, these results should be interpreted as hypothesis generators to narrow down further developmental and functional studies to deepen our understanding about the nervous system evolution and sensory system rearrangements occurring in cave-dwelling beetles'*.

Reviewer #3 (Remarks to the Author):

My concerns have been addressed in the revisions.

> Authors: We thank the reviewer for their constructive comments on the initial version of the manuscript, which have helped to substantially improve it.

Reviewers' Comments:

Reviewer #2:

Remarks to the Author:

I congratulate the authors on addressing my comments diligently and successfully.